# Neutrophils promote the development of reparative macrophages mediated by ROS to orchestrate liver repair

Wenting Yang[1], Yuandong Tao[1], Yan Wu[1], Xinyuan Zhao[1], Weijie Ye[1], Dianyuan Zhao[1], Ling Fu[1], Caiping Tian[1], Jing Yang [1], Fuchu He[1] & Li Tang[1,2]

Phagocytes, including neutrophils and macrophages, have been suggested to function in a cooperative way in the initial phase of inflammatory responses, but their interaction and integration in the resolution of inflammation and tissue repair remain unclear. Here we show that neutrophils have crucial functions in liver repair by promoting the phenotypic conversion of pro-inflammatory Ly6C$^{hi}$CX$_3$CR1$^{lo}$ monocytes/macrophages to pro-resolving Ly6C$^{lo}$CX$_3$CR1$^{hi}$ macrophages. Intriguingly, reactive oxygen species (ROS), expressed predominantly by neutrophils, are important mediators that trigger this phenotypic conversion to promote liver repair. Moreover, this conversion is prevented by the depletion of neutrophils via anti-Ly6G antibody, genetic deficiency of granulocyte colony-stimulating factor, or genetic deficiency of NADPH oxidase 2 (Nox2). By contrast, adoptive transfer of WT rather than Nox2$^{-/-}$ neutrophils rescues the impaired phenotypic conversion of macrophages in neutrophil-depleted mice. Our findings thus identify an intricate cooperation between neutrophils and macrophages that orchestrate resolution of inflammation and tissue repair.

---

[1] State Key Laboratory of Proteomics, National Center for Protein Sciences, Beijing. Beijing Proteome Research Center, Beijing Institute of Lifeomics, 102206 Beijing, China. [2] Department of Biochemistry and Molecular Biology, Anhui Medical University, 230032 Hefei, Anhui, China. These authors contributed equally: Wenting Yang, Yuandong Tao. Correspondence and requests for materials should be addressed to F.H. (email: hefc@nic.bmi.ac.cn) or to L.T. (email: tangli08@aliyun.com)

Neutrophils are first responders that traffic from circulation to the site of injury[1]. Although they are generally thought to exacerbate tissue injury through releasing proteases and oxidants, recent work has implicated that neutrophils may exhibit anti-inflammatory or healing characteristics[2,3]. Neutrophils have been shown to generate a number of important anti-inflammatory and pro-resolving lipid mediators, which program the end of inflammation[4,5]. Furthermore, neutrophils can possess immunosuppressive functions and contribute to host protection in various contexts, including experimental colitis, rheumatoid arthritis, and endotoxemia[6–8]. Additionally, using intravital microscopy, recent studies have shown that tissue-infiltrated neutrophils perform key repair functions and finally migrate back into the bone marrow[9]. Although the role of neutrophils in the regulation of inflammation and damage repair has been increasingly appreciated, the mechanisms by which neutrophils contribute to the resolution of inflammation remain largely unexplained. And whether they coordinate with surrounding cell types to trigger the resolution program is incompletely defined.

In contrast to neutrophils, the proactive role of macrophages in tissue repair has been extensively described in the last few years[10]. Shortly after the extravasation of neutrophils into an inflamed tissue, blood monocytes are abundantly recruited, which then differentiate into macrophages and dendritic cells (DCs)[11]. Accumulating evidence suggests that monocyte-derived macrophages can undergo phenotypic and functional transition for effective wound healing and tissue regeneration[12,13]. Commonly, pro-inflammatory $Ly6C^{hi}CX_3CR1^{lo}$ monocytes/macrophages can convert, in situ, to anti-inflammatory or reparative $Ly6C^{lo}CX_3CR1^{hi}$ macrophages[13]. Local tissue signals are thought to have considerable influence on the reprograming of macrophages[14]. However, the cell types within the local microenvironment and the molecular mechanisms that instruct macrophages to adopt a repair phenotype are not fully understood.

Acetaminophen (N-acetyl-p-aminophenol, APAP) overdose can cause severe liver injury and is a leading cause of acute liver failure (ALF) in many developed countries[15]. Multiple studies have demonstrated that a substantial number of innate immune cells, especially neutrophils and monocyte-derived macrophages, accumulate in inflamed livers[16,17]. Here, we use a murine model of APAP-induced liver injury to study the coordinated orchestration of diverse inflammatory cellular components in the process of inflammation resolution. We show that neutrophils instruct, potentially via ROS, inflammatory monocytes/macrophages to adopt a pro-regenerative phenotype for optimal liver repair. Our findings thus identify a previously unappreciated neutrophil-macrophage interaction that facilitates liver regeneration and repair, and uncover how phagocytic populations may be an integral part of fine-tuning tissue repair.

## Results

**Neutrophils contribute to liver repair after acute injury**. APAP-induced liver injury displayed distinct injury (0–24 h) and resolution (48–72 h) phases[18]. In accordance with previous reports[13], we found that monocytes/macrophages and neutrophils were the major cell types among the infiltrating cells (Supplementary Fig. 1a). Although $Ly6C^{hi}CX_3CR1^{lo}$ cells are often termed inflammatory monocytes, we found that they also expressed macrophage markers (F4/80, CD64, and the tyrosine kinase MerTK) (Supplementary Fig. 1b), suggesting that this cell population may include infiltrating monocytes, monocytes in differentiation to macrophages and/or macrophages. Thus, we identified them as $Ly6C^{hi}CX_3CR1^{lo}$ monocytes/macrophages here. In addition, consistent with previous observations[19], we

found that monocyte-deficient $Ccr2^{-/-}$ mice exhibited impaired resolution of hepatic damage (Supplementary Fig. 1c–e), suggesting that monocyte-derived macrophages play an essential role in liver repair.

To study the importance of neutrophils for liver repair, we depleted neutrophils by in vivo administration of antibodies specific for Ly6G (Fig. 1a and Supplementary Fig. 2a). Neutrophil depletion resulted in higher serum ALT levels during the resolution phase (Fig. 1b). Histological analysis revealed increased necrotic areas and lower numbers of Ki67+ hepatocytes at 72 h in the livers of neutrophil-depleted mice (Fig. 1c, d). Because hepatocyte morphology is different from other non-parenchymal cells, we selected cells with brown stained large nuclei as Ki67-positive hepatocytes for quantification, as previously described[20]. In an alternative strategy for neutrophil depletion, we took advantage of granulocyte colony-stimulating factor deficient mice ($Gcsf^{-/-}$). Consistent with previous reports[21], $Gcsf^{-/-}$ mice exhibited a significant reduction in circulating neutrophils and a relatively small reduction in circulating monocytes (Supplementary Fig. 2b). Notably, genetic neutropenia similarly exhibited exacerbated injury and diminished liver regeneration compared with WT mice during the resolution phase (Fig. 1e–g). Thus, these results suggest that neutrophils contribute to liver repair after acute liver injury.

**Neutrophils mediate development of reparative macrophages**. Unlike macrophages, the number of neutrophils progressively decreased during the resolution phase, which suggests that it is unlikely that they directly mediate the resolution of inflammation but raises the possibility that they contribute to liver repair indirectly by providing signals for other reparative cells, such as macrophages. We observed that neutrophil depletion resulted in a remarkable increase in $Ly6C^{hi}CX_3CR1^{lo}$ monocytes/macrophages and a significant reduction in $Ly6C^{lo}CX_3CR1^{hi}$ macrophages at 72 h after APAP injection (Fig. 2a). Moreover, neutrophil ablation induced a decrease of pro-resolving marker (Hgf and Mrc1) and an increase of pro-inflammatory marker expression (Il1b) in 72 h monocyte-derived macrophages (Fig. 2b). Hepatocyte growth factor (HGF) is known to mediate liver regeneration[22] and Macrophage mannose receptor (Mrc1) is considered as a marker for alternatively activated or M2 macrophages[23].

Because pro-inflammatory $Ly6C^{hi}CX_3CR1^{lo}$ monocytes/macrophages can convert to reparative $Ly6C^{lo}CX_3CR1^{hi}$ macrophages during the resolution phase[13,24], we hypothesized that neutrophils might mediate the phenotypic switch of macrophages to facilitate proper repair. Indeed, we found that the engrafted $Ly6C^{hi}$ monocytes switched into $Ly6C^{lo}$ macrophages in the inflamed liver of recipient mice (Supplementary Fig. 3a-c), as previously reported[12,13]. Next, we wanted to examine whether neutrophils mediate macrophage skewing in vivo. We transferred CD45.1+Ly6C$^{hi}$ monocytes into APAP-challenged CD45.2 recipient mice that had been depleted of neutrophils (Fig. 2c, Supplementary Fig. 4a). Significant lower proportion of the $Ly6C^{hi}$ monocytes converted to the $Ly6C^{lo}$ macrophages in the neutrophil-depleted recipients compared with the control recipients (Fig. 2d). Thus, these data demonstrate a requirement for neutrophils in macrophage skewing.

To further confirm the role of neutrophils in vitro, we prepared a neutrophil and $Ly6C^{hi}CX_3CR1^{lo}$ monocyte/macrophage co-culture system (Fig. 2e, Supplementary Fig. 4b). Considering purified neutrophil life span is limited, we evaluated neutrophil viability before co-culture with macrophages and found that the proportion of living neutrophils was over 95% (Supplementary Fig. 5a). Conditioned medium from 24 h hepatic neutrophils significantly upregulated the expression of wound healing-related

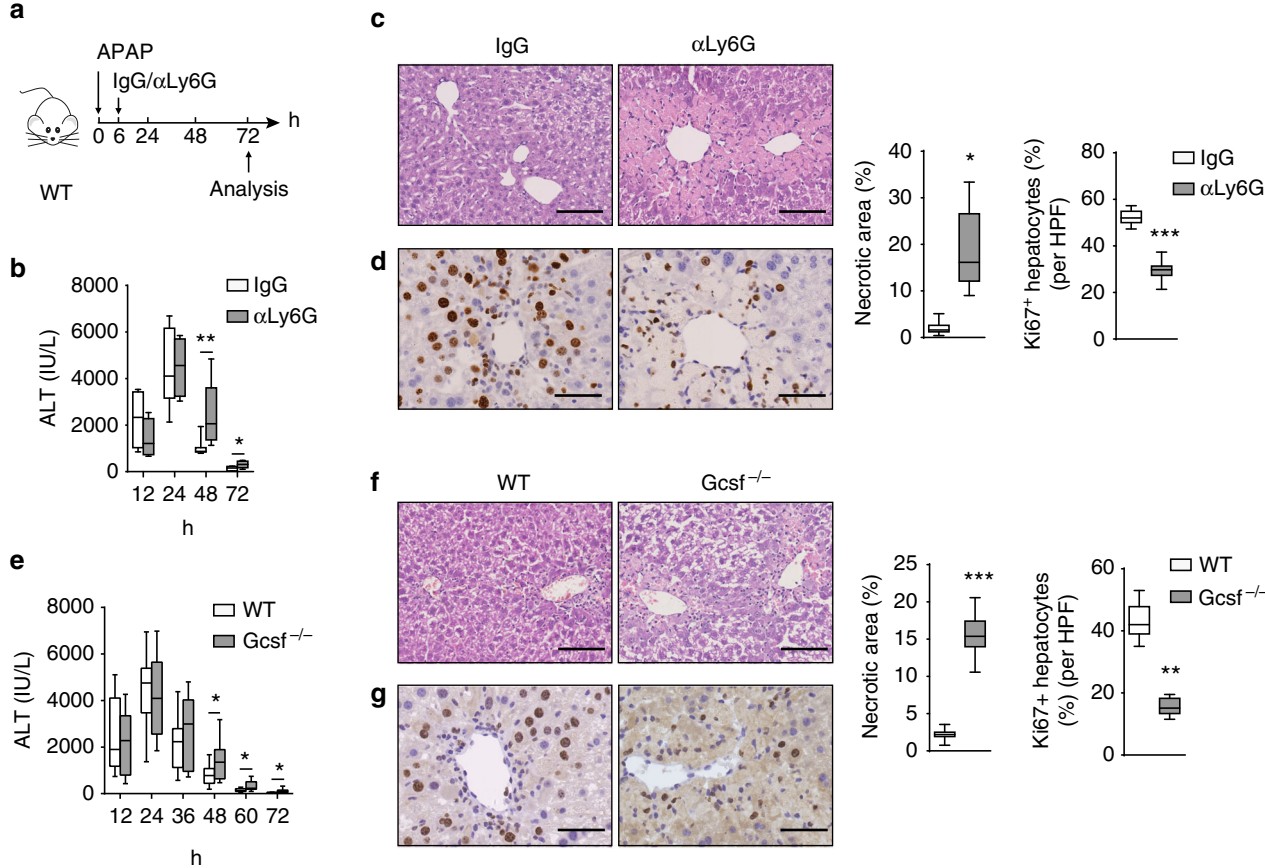

**Fig. 1** Neutrophils contribute to liver repair after acute injury. **a–d** Schematic of the experimental design. **a** Mice were treated with the neutrophil-depleting anti-Ly6G mAb or an isotype control (IgG) at 6 h after APAP challenge. Serum ALT levels at each time point (**b**), histological characterization (**c**), and IHC staining for Ki67 (**d**) in liver sections at 72 h were evaluated. Percentage of necrotic areas and Ki67 positive hepatocytes per high-powered field (HPF) were also quantified. $n = 5$. Experiment was repeated three times. **e–g** WT and Gcsf$^{-/-}$ mice were challenged with APAP. Serum ALT levels at each time point (**e**), histological characterization (**f**), and IHC staining for Ki67 (**g**) in liver sections at 72 h are shown. $n = 13, 12$. The data are pooled from two independent experiments. The bar graph indicates 100 μm (**c**, **f**), 50 μm (**d**, **g**). Whiskers show min to max. Bars show the median (*$P < 0.05$, **$P < 0.01$, ***$P < 0.001$). $P$ values were calculated by two-tailed Student's $t$-test (**b–g**)

genes accompanied by the downregulated expression of the pro-inflammatory-related gene in Ly6C$^{hi}$CX$_3$CR1$^{lo}$ monocytes/macrophages (Fig. 2f), suggesting a skewing toward the pro-resolving phenotype. Moreover, conditioned medium from neutrophil-treated Ly6C$^{hi}$CX$_3$CR1$^{lo}$ monocytes/macrophages, but not neutrophils or the Ly6C$^{hi}$CX$_3$CR1$^{lo}$ monocytes/macrophages medium alone, was able to induce hepatocyte proliferation (Fig. 2g, h). Thus, these observations indicate that neutrophils trigger a phenotypic switch from pro-inflammatory Ly6C$^{hi}$CX$_3$CR1$^{lo}$ monocytes/macrophages to pro-resolving Ly6C$^{lo}$CX$_3$CR1$^{hi}$ macrophages and then accelerate liver repair.

**ROS are required for liver repair and macrophage skewing.** Because activated neutrophils can potently produce ROS and phagocyte NADPH oxidase 2 complex-derived ROS are known to be crucial regulators of immune response[25,26], we asked whether ROS might mediate the regulation of the macrophage skewing. Using the fluorescent probe CM-H$_2$DCFDA to detect intracellular levels of ROS, we observed that neutrophils were the predominant ROS-producing cell population among the analyzed immune cell types in the livers of APAP-challenged mice (Fig. 3a and Supplementary Fig. 5b). In addition, serum and liver homogenates contained significant amounts of hydrogen peroxide (H$_2$O$_2$) after APAP challenge (Fig. 3b); the amounts of

catalase, a hydrogen peroxide degrading enzyme, were comparable in liver homogenates between normal and APAP-challenged mice (Supplementary Fig. 5c). Moreover, higher amounts of H$_2$O$_2$ were detected in the supernatants of hepatic neutrophils from APAP-challenged mice compared to hepatic macrophages or non-activated neutrophils and monocytes from the control mice (Fig. 3c). Thus, neutrophils serve as a major source of ROS in APAP-induced liver injury.

We then asked whether ROS could influence the resolution of inflammation in vivo. Because ROS are mainly produced by the phagocytic Nox2 complex[27], we explored the role of ROS using Nox2$^{-/-}$ mice, in which the gp91 subunit of NADPH oxidase is deleted. Nox2$^{-/-}$ mice displayed delayed resolution of liver injury, as evidenced by the drastic increase in ALT levels, persistence of necrotic debris, and a substantial reduction in the number of Ki67$^+$ hepatocytes during the resolution phase (Fig. 3d–f). Additionally, Nox2$^{-/-}$ mice exhibited a remarkable increase in the number of pro-inflammatory Ly6C$^{hi}$CX$_3$CR1$^{lo}$ monocytes/macrophages and a significant reduction in the number of reparative Ly6C$^{lo}$CX$_3$CR1$^{hi}$ macrophages (Fig. 3g). Furthermore, Nox2$^{-/-}$ mice also resulted in a decrease of pro-resolving marker and an increase of pro-inflammatory marker expression in 72 h monocyte-derived macrophages (Fig. 3h). We next examined the possibility that the impaired liver repair was

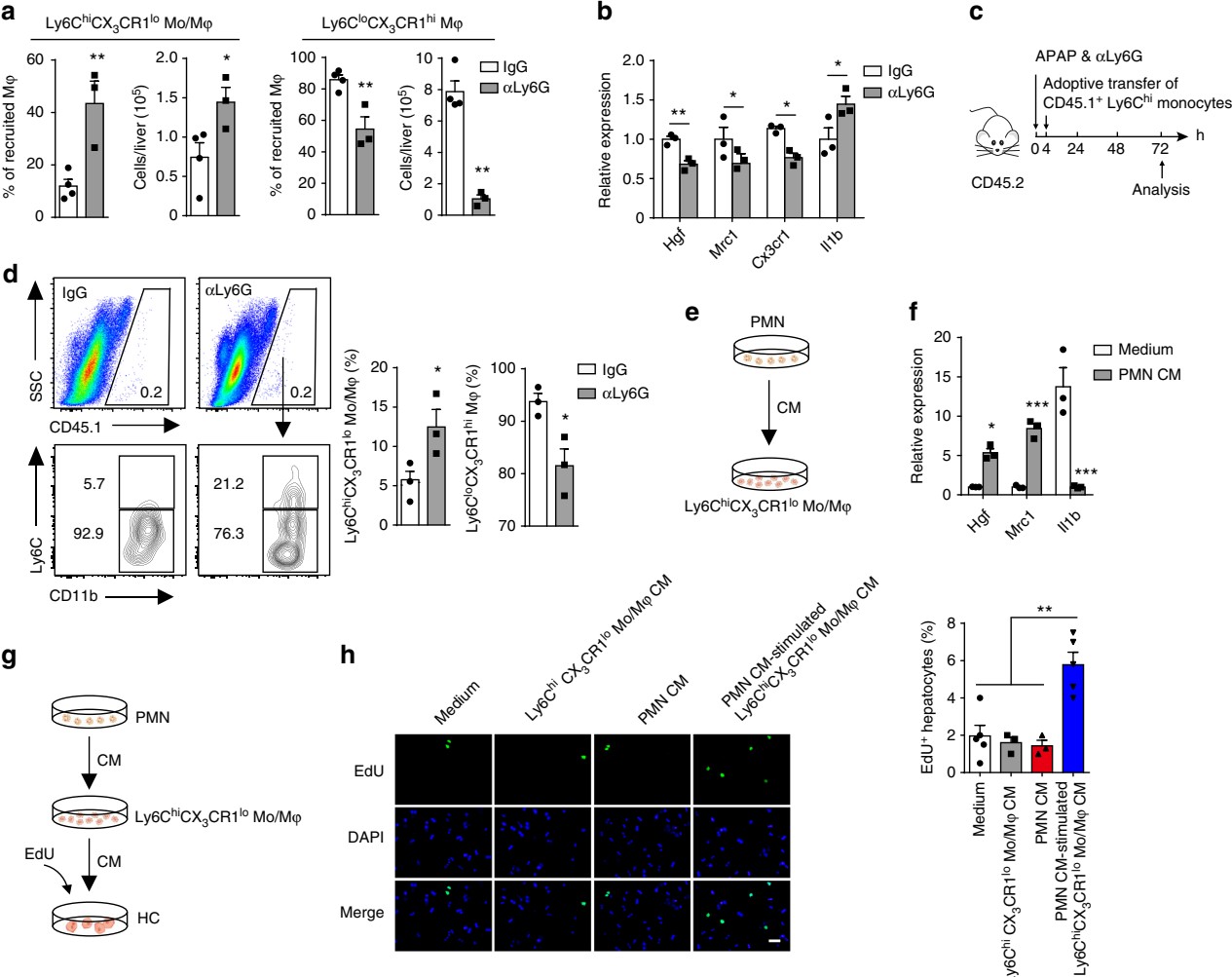

**Fig. 2** Neutrophils mediate development of reparative macrophages. **a, b** Mice were treated with anti-Ly6G mAb or IgG at 6 h after APAP challenge. The percentage and number of the indicated macrophage subsets at 72 h were determined (**a**). $n = 4, 3$. Experiment was repeated three times. Gene expression in 72 h hepatic monocyte-derived macrophages following neutrophil depletion was measured by qPCR, presented relative to *Gapdh* (**b**). $n = 3$. Experiment was repeated three times. **c, d** Schematic of the experimental design: CD115+Ly6Chi monocytes enriched with CD45.1 mice were adoptively transferred into neutrophil-depleted CD45.2 mice at 4 h after APAP challenge (**c**). The transferred CD45.1+ monocytes in the livers of recipient mice harvested at 72 h were identified (**d**). $n = 3$. Experiment was repeated twice. **e, f** Schematic of the experimental design: 24 h Ly6ChiCX3CR1lo monocytes/macrophages were exposed to the conditioned medium (CM) of 24 h hepatic neutrophils (**e**). The expression of the indicated genes in Ly6ChiCX3CR1lo monocytes/macrophages was detected by qPCR (**f**). $n = 3$. Experiment was repeated three times. **g, h** Schematic of the experimental design: Hepatocyte proliferation was induced with the supernatants from 24 h Ly6ChiCX3CR1lo monocytes/macrophages that had been co-cultured with the CM of 24 h neutrophils compared to the supernatants from the 24 h Ly6ChiCX3CR1lo monocytes/macrophages or neutrophils cultured alone (**g**). Hepatocytes undergoing DNA synthesis after co-culture were visualized using a EdU Imaging Kit. The EdU-positive cells were then quantified (**h**). The bar graph indicates 50 μm. $n = 5, 3, 3, 5$. Experiment was repeated twice. The results represent mean ± s.e.m. (*$P < 0.05$, **$P < 0.01$, ***$P < 0.001$). $P$ values were calculated by two-tailed Student's $t$-test (**a**, **b**, **d**, **f**) and one-way ANOVA (**h**)

caused by alterations of APAP metabolism[28]. Hepatic glutathion (GSH) levels were measured at various time points after APAP challenge. There was no significant difference in hepatic GSH concentrations between WT and Nox2−/− mice (Supplementary Fig. 5d), suggesting that the delayed liver repair in Nox2−/− mice may not be due to altered APAP metabolism.

We further tested whether ROS blockade affects the conversion of Ly6ChiCX3CR1lo monocytes/macrophages into Ly6CloCX3CR1hi macrophages. Since monocytes express CX3CR1 and can be tracked in *Cx3cr1CreERT2-EYFP* mice[29,30], we traced the fate of *Cx3cr1+* monocyte-derived macrophages by crossing *Cx3cr1CreERT2-EYFP* mice with *Rosa26tdTomato* reporter mice. *Cx3cr1CreERT2-EYFP/+Rosa26tdTomato/+* mice were administrated with tamoxifen, followed

1 day later by APAP treatment. At 24 h after APAP injection, 86% of hepatic CD11b+tdTomato+ macrophages were Ly6Chi, whereas at 72 h after APAP challenge 91% of hepatic CD11b+tdTomato+ macrophages were Ly6Clo, suggesting a conversion from Ly6ChiCX3CR1lo monocytes/macrophages toward Ly6CloCX3CR1hi macrophages (Supplementary Fig. 3d, e). To further confirm the role of ROS in macrophage skewing, we took advantage of N-acetylcysteine (NAC), which is a commonly used ROS scavenger. We blocked ROS by NAC in *Cx3cr1CreERT2-EYFP/+Rosa26tdTomato/+* mice that had been treated with tamoxifen and APAP (Fig. 3i). At 72 h post APAP challenge, 85% of tdTomato+ macrophages were Ly6Clo in PBS-treated mice, compared with 57% in NAC-treated mice, which indicated impaired macrophage conversion (Fig. 3j,

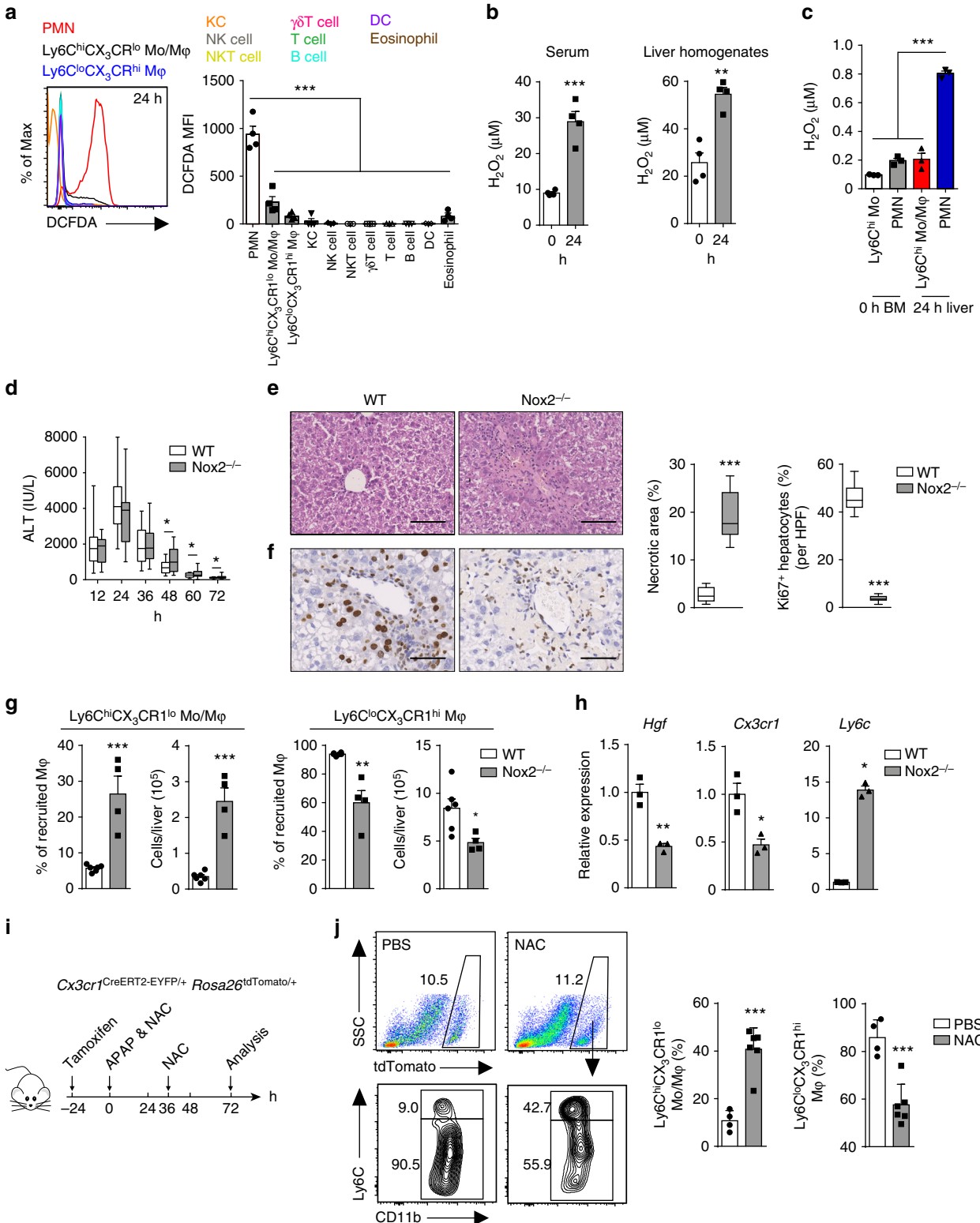

Supplementary Fig. 4c). Together, these data suggest that ROS are required for optimal liver repair by promoting macrophage skewing toward a reparative phenotype. It is worth noting that although NAC also acts as an antidote in APAP intoxication, the effect of NAC we observed during the resolution phase may be independent of its detoxification mechanism. There are two main reasons: Firstly, the detoxification mechanism of NAC is only effective during the early

phase following APAP overdose in humans and mice[31–33]. Secondly, Late or prolonged treatment with NAC is indeed detrimental rather than protective in mice[34,35], which could not be explained by APAP detoxification.

**ROS-producing neutrophils permit macrophage skewing.** To further investigate the importance of ROS produced by

**Fig. 3** ROS facilitate liver repair through regulating macrophage skewing in vivo. **a** Flow cytometric analysis of CM-H$_2$DCFDA staining for ROS in the indicated cell types at 24 h after APAP challenge. MFI, mean fluorescence intensity. $n = 4$. Experiment was repeated three times. **b** The levels of H$_2$O$_2$ in serum and liver homogenates were determined. $n = 4$. Experiment was repeated twice. **c** The extracellular ROS concentrations from Ly6C$^{hi}$ monocytes (Mo) and neutrophils in the bone marrow (BM) of untreated mice, and hepatic 24 h Ly6C$^{hi}$CX$_3$CR1$^{lo}$ monocytes/macrophages and neutrophils of APAP-treated mice were measured. $n = 3$. Experiment was repeated three times. **d–f** WT and Nox2$^{-/-}$ mice were challenged with APAP. Serum ALT levels at each time point (**d**), H&E staining (**e**), and IHC staining for Ki67 (**f**) in liver sections at 72 h were evaluated. $n = 17, 21$. The data shown are pooled from three independent experiments. Whiskers show min to max. Bars show the median. **g, h** The percentage and number of the indicated macrophage subsets among the total hepatic recruited macrophages at 72 h were determined (**g**). $n = 6, 4$. Gene expression in 72 h hepatic monocyte-derived macrophages was measured by qPCR (**h**). $n = 3$. Experiment was repeated three times. **i, j** Schematic of the experimental design: Cx3cr1$^{CreERT2-EYFP/+}$Rosa26$^{tdTomato/+}$ mice were given tamoxifen via oral gavage 1 day before APAP challenge. Cx3cr1$^{CreERT2-EYFP/+}$Rosa26$^{tdTomato/+}$ mice also received two doses of NAC with a separation of 36 h between doses to block ROS (**i**). Representative flow cytometry analysis of Ly6C expression on CD11b$^+$tdTomato$^+$ cells in the livers of PBS- or NAC-treated mice at 72 h post APAP challenge (**j**). $n = 4, 6$. Experiment was repeated twice. The bar graph indicates 100 μm (**e**), 50 μm (**f**). The data shown are mean ± s.e.m. (*$P < 0.05$, **$P < 0.01$, ***$P < 0.001$). $P$ values were calculated by one-way ANOVA (**a**, **c**) and two-tailed Student's $t$-test (**b**, **d–h**, **j**)

neutrophils in liver repair, we transplanted WT recipient mice with a 50/50 mixture of bone marrow cells from Nox2$^{-/-}$ and Gcsf$^{-/-}$ mice or WT and Gcsf$^{-/-}$ mice (Fig. 4a). Because mice lacking G-CSF have chronic neutropenia and impaired neutrophil mobilization[21,36], these bone marrow chimeras contain primarily WT or Nox2$^{-/-}$ neutrophils. Liver repair was markedly impaired in the mixed bone marrow chimeras in which Nox2-derived ROS were deficient in neutrophils compared to the WT controls (Fig. 4b–d and Supplementary Fig. 5e). Moreover, the number of pro-inflammatory Ly6C$^{hi}$CX$_3$CR1$^{lo}$ monocytes/macrophages was significantly increased, while the number of reparative Ly6C$^{lo}$CX$_3$CR1$^{hi}$ macrophages was markedly decreased in the Nox2$^{-/-}$/Gcsf$^{-/-}$ mice compared to the WT/Gcsf$^{-/-}$ mice (Fig. 4e).

Furthermore, we performed an adoptive transfer experiment to confirm the role of ROS produced by neutrophils. We adoptively transferred purified neutrophils derived from bone marrow of WT or Nox2$^{-/-}$ donors into APAP-challenged neutrophil-depleted recipients (Fig. 4f, Supplementary Fig. 6a). In both groups, transferred neutrophils were detected in the blood and liver of recipient mice (Supplementary Fig. 6b). Notably, adoptive transfer of WT neutrophils rescued the exacerbated damage and depressed hepatic regeneration in neutrophil-depleted mice during the resolution phase, but transfer of Nox2$^{-/-}$ neutrophils failed to do so (Fig. 4g–i). Moreover, transfer of WT but not Nox2-deficient neutrophils promoted macrophage conversion in neutrophil-depleted mice (Fig. 4j). Therefore, ROS-producing neutrophils are instrumental for optimal liver repair. However, we cannot exclude that additional factors other than ROS might contribute to macrophage skewing and liver repair. Since neutrophils are reported to generate many anti-inflammatory and pro-resolving mediators[5], whether these products have similar effects remains to be investigated. Nevertheless, we identified ROS expressed predominantly by neutrophils are important regulators of macrophage skewing for liver repair.

**ROS mediate macrophage skewing ex vivo**. Having uncovered that ROS-producing neutrophils mediate the resolution of inflammation in vivo, we examined whether ROS could confer a pro-resolving phenotype on pro-inflammatory macrophages ex vivo. We took advantage of the neutrophil-macrophage co-culture system again. Neutrophil-conditioned medium promoted pro-inflammatory Ly6C$^{hi}$CX$_3$CR1$^{lo}$ monocytes/macrophages skewing toward a reparative phenotype, but the addition of cat-alase induced a decrease of pro-resolving marker and an increase of pro-inflammatory marker expression (Fig. 5a). Similarly, conditioned medium from Nox2$^{-/-}$ neutrophils failed to upre-gulate Hgf or reduce Il1b and Ly6c in the Ly6C$^{hi}$CX$_3$CR1$^{lo}$

monocytes/macrophages (Fig. 5b), indicating a requirement of ROS released by neutrophils in educating macrophages to a reparative phenotype. Furthermore, conditioned medium from the WT neutrophil-treated Ly6C$^{hi}$CX$_3$CR1$^{lo}$ monocytes/macrophages significantly stimulated hepatocyte DNA replication, whereas conditioned medium from the Nox2$^{-/-}$ neutrophil-treated Ly6C$^{hi}$CX$_3$CR1$^{lo}$ monocytes/macrophages failed to induce hepatocyte proliferation (Fig. 5c). Collectively, these results strongly suggest that ROS released by neutrophils mediate macrophage skewing toward a reparative phenotype, thus leading to hepatocyte proliferation. Considering that ROS can diffuse across membranes and react with other cells in the vicinity[37], the ROS produced by neutrophils are easily capable of regulating macrophage phenotype via the paracrine mechanism.

Next, we performed transcriptomal profiling of macrophages after exogenous addition of H$_2$O$_2$. Peritoneal macrophages were polarized into a pro-inflammatory phenotype with lipopolysac-charide (LPS) and then were stimulated with 20 μM H$_2$O$_2$ for 4 h. We observed that H$_2$O$_2$-treated peritoneal macrophages displayed increased expression of a large number of wound healing genes and some M2 genes (Fig. 5d). Meanwhile, a number of pro-inflammatory genes or M1 genes were down-regulated in H$_2$O$_2$-treated peritoneal macrophages (Fig. 5d). qPCR analysis confirmed these results by the addition of 20 μM H$_2$O$_2$ (Fig. 5e). Note that this concentration was not detrimental for the cells within 4 h of exposure. These findings reveal that ROS can induce a tissue-repair gene-expression program in macrophages.

**ROS promote activation of AMPK in macrophages**. To obtain insight into the molecular mechanism by which ROS trigger macrophage skewing, we screened the activation status of various kinases or transcription factors that translate signals into a polarized macrophage phenotype[23,38]. H$_2$O$_2$ treatment at the indicated concentrations had no significant effects on the acti-vation of JAK1, JAK2, STAT1, STAT3, STAT6, ERK1/2, and NF-κB at the time points observed (Fig. 6a). However, activation of AMP-activated protein kinase (AMPK) was significantly enhanced in both peritoneal macrophages and bone marrow-derived macrophages (BMDMs) after exposure to H$_2$O$_2$ (Fig. 6b). AMPK, a master regulator of energy homeostasis, has been reported to enhance alternative macrophage activation and reduce macrophage-mediated inflammation[39,40]. In addition, ROS have been shown to activate AMPK in macrophages, neu-trophils and other cell populations[41,42]. We thus hypothesized that ROS might regulate macrophage skewing by activating AMPK. Since ROS can modulate the function of intracellular signaling pathways through oxidative modification of specific

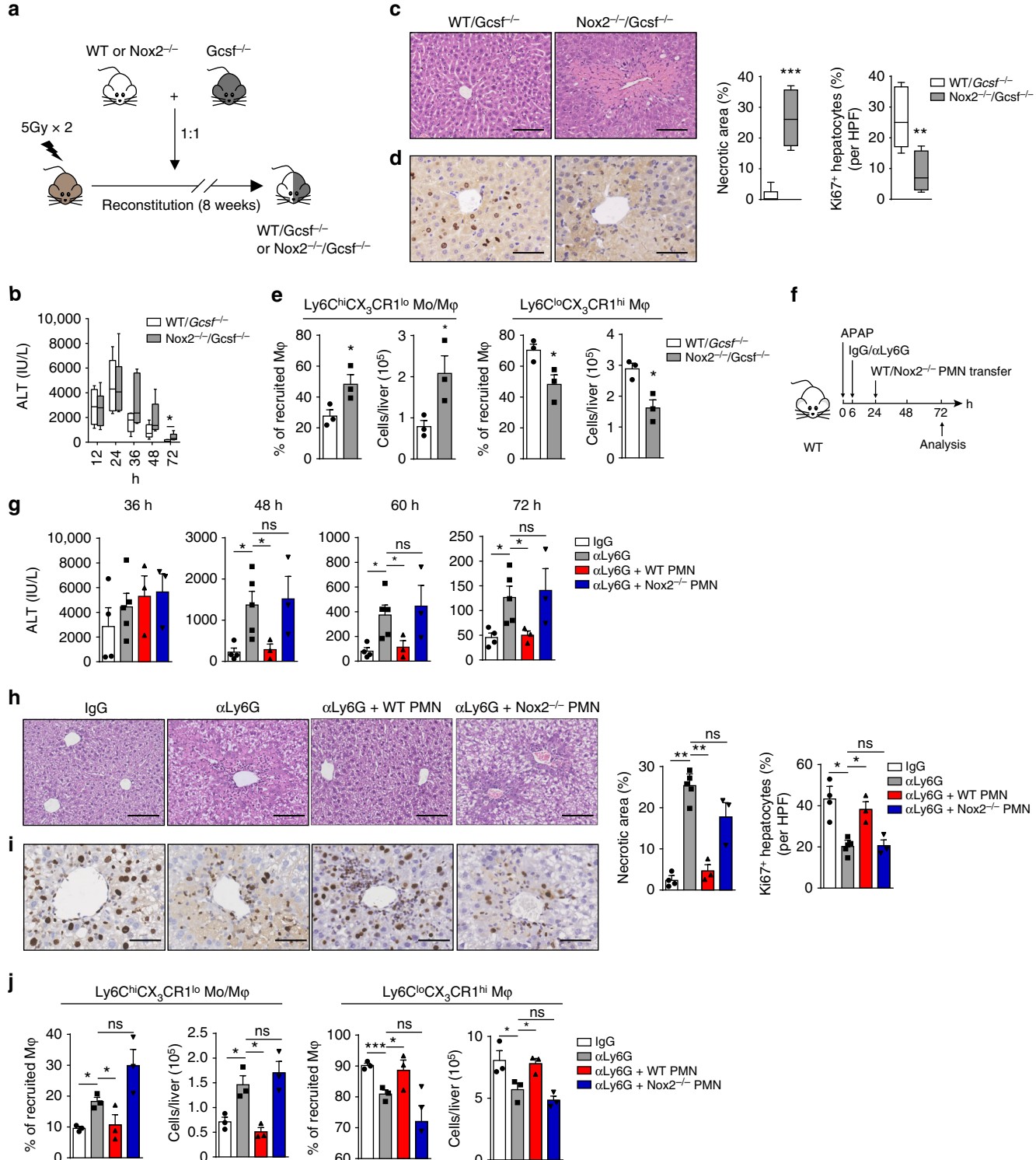

**Fig. 4** ROS-producing neutrophils permit macrophage skewing toward a reparative phenotype. **a–e** Schematic of the experimental design: mixed bone marrow chimera mice were generated by reconstituting lethally irradiated WT recipients with 50% WT + 50% Gcsf$^{-/-}$ or 50% Nox2$^{-/-}$ + 50% Gcsf$^{-/-}$ bone marrow cells (total transplanted cells were 10$^7$ cells per mouse) (**a**). Chimera mice were challenged with APAP. Serum ALT levels at the indicated time points (**b**), histological characterization (**c**), and IHC staining for Ki67 (**d**) in liver sections at 72 h were evaluated. $n = 6$. Whiskers show min to max. Bars show the median. The percentage and number of the indicated macrophage subsets at 72 h were calculated (**e**). $n = 3$. Experiment was repeated twice. **f–j** Schematic of the experimental design: WT mice were treated with neutrophil-depleting anti-Ly6G mAb at 6 h after APAP challenge; WT or Nox2$^{-/-}$ neutrophils were adoptively transferred to the anti-Ly6G-treated mice at 24 h post APAP challenge (**f**). Serum ALT levels at the indicated time points (**g**), histological characterization (**h**), and IHC staining for Ki67 (**i**) in liver sections at 72 h were evaluated. $n = 4, 5, 3, 3$. The percentage and number of the indicated macrophage subsets at 72 h were calculated (**j**). $n = 3$. Experiment was repeated twice. The bar graph indicates 100 μm (**c**, **h**), 50 μm (**d**, **i**). The data shown are mean ± s.e.m. (*$P < 0.05$, **$P < 0.01$, ***$P < 0.001$). $P$ values were calculated by two-tailed Student's $t$-test (**c–e**, **g–j**)

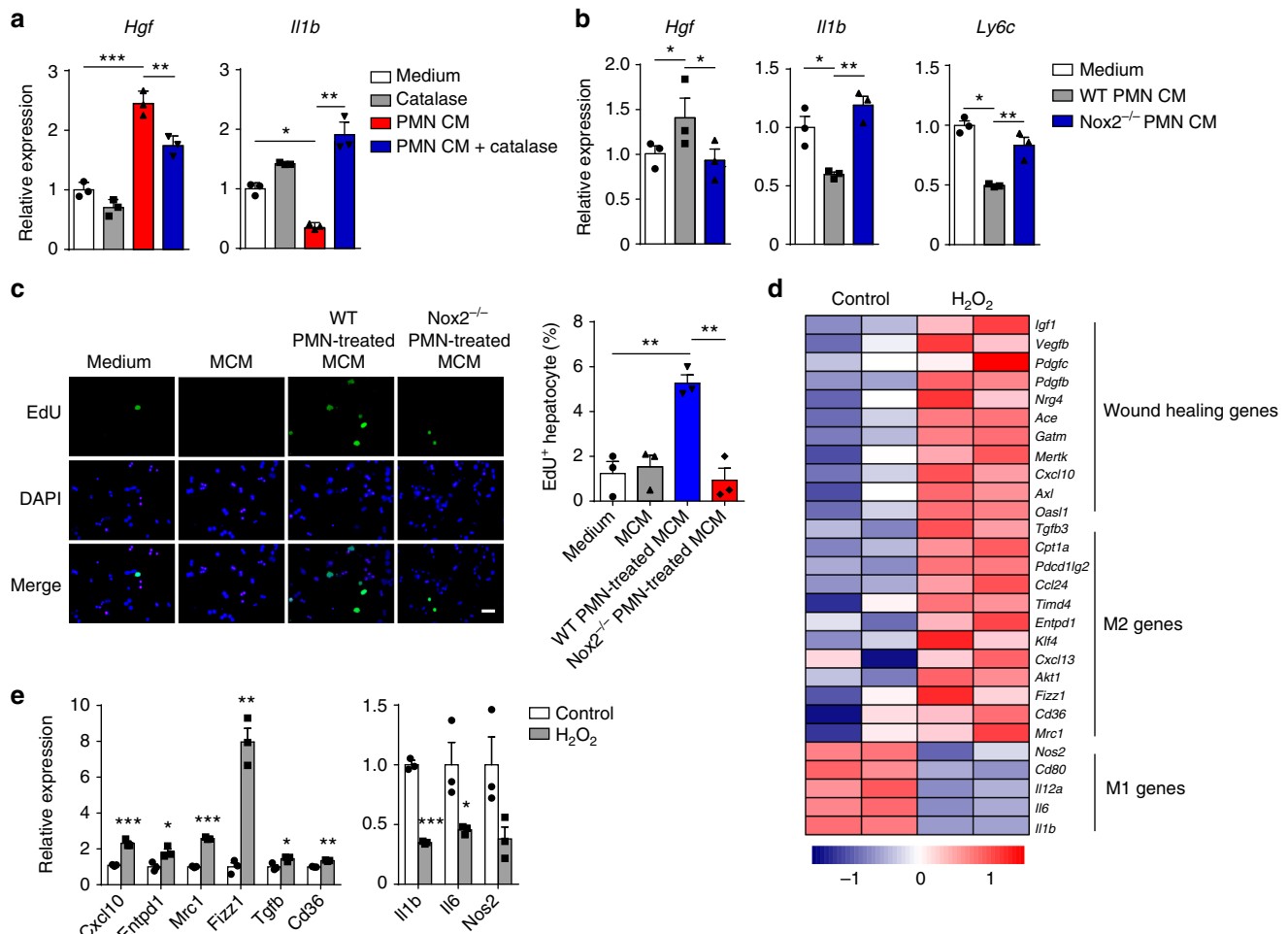

**Fig. 5** ROS mediate phenotypic conversion of pro-inflammatory macrophages ex vivo. **a** The 24 h Ly6C^hiCX_3CR1^lo monocytes/macrophages were co-cultured with CM from 24 h neutrophils in the absence or presence of catalase for 6 h. The expression of the indicated genes was measured by qPCR. $n = 3$. Experiment was repeated twice. **b** The 24 h Ly6C^hiCX_3CR1^lo monocytes/macrophages were co-cultured with CM from 24 h WT or Nox2^−/− neutrophils for 6 h. The expression of the indicated genes was measured by qPCR. $n = 3$. Experiment was repeated twice. **c** Hepatocyte proliferation was induced by the CM from the 24 h Ly6C^hiCX_3CR1^lo monocytes/macrophages (MCM) that had been exposed to CM from 24 h WT or Nox2^−/− neutrophils. Representative images of hepatocytes pulsed with EdU (left panel) and the quantification of hepatocyte proliferation (right panel). Scale bar, 50 μm. $n = 3$. Experiment was repeated twice. **d** Peritoneal macrophages were treated with LPS (100 ng ml^−1) for 3 h to polarize into pro-inflammatory phenotype and then were stimulated with 20 μM H_2O_2 for 4 h. Heatmap showing differential expression of the indicated genes in H_2O_2-treated macrophages as compared to control macrophages. Row-based Z-score normalized. **e** Expression of the indicated genes in H_2O_2-treated or untreated macrophages were determined by qPCR. $n = 3$. Experiment was repeated twice. The results represent mean ± s.e.m. (*$P < 0.05$, **$P < 0.01$, ***$P < 0.001$). $P$ values were calculated by two-tailed Student's $t$-test (**a**, **b**, **e**) and one-way ANOVA (**c**)

cysteine residues within proteins[43], we wondered whether exposure to H_2O_2 could lead to direct oxidation of AMPK. We examined the oxidation of AMPK by biotinylated iodoacetamide (BIAM) labeling assay, which could determine the extent of free (unoxidized) cysteine residues within target proteins[44]. However, we did not observe a decrease in BIAM-protein adduct formation, indicating that there was no increased oxidation of cysteine residues within AMPK after H_2O_2 treatment (Fig. 6c). Under the same conditions, we detected increased oxidation of Peroxiredoxin-1 (Prdx1), which was a sensor of intracellular oxidants[45]. These results indicate that H_2O_2 may promote AMPK activation by regulating its upstream kinases rather than directly oxidizing AMPK itself.

AMPK activation can be triggered by upstream Ca^{2+}-CaMKKβ pathway[41,46]. Stimulation of macrophages with different concentrations of H_2O_2 led to immediate Ca^{2+} influx, and the activation of AMPK was markedly reduced by the removal of extracellular

Ca^{2+} (Fig. 6d, e). Furthermore, inhibition of CaMKKβ activity by STO-609 also suppressed the phosphorylation of AMPK (Fig. 6f), suggesting a role for Ca^{2+}-CaMKKβ pathway in ROS-induced AMPK activation. Meanwhile, we observed different effects of Ca^{2+} removal and CaMKKβ inhibition on AMPK activation in untreated macrophages (Fig. 6e, f), which might be due to Ca^{2+} independent activity of CaMKKβ.

Next, we investigated the phosphorylation of AMPK after APAP challenge in vivo. Using in situ fluorescence labeling of p-AMPK and F4/80, we observed that higher numbers of 72 h macrophages expressed p-AMPK in comparison to normal or 24 h macrophages (Supplementary Fig. 7a). In addition, increased phosphorylation of AMPK was observed in 72 h Ly6C^loCX_3CR1^hi macrophages compared to 24 h Ly6C^hiCX_3CR1^lo monocytes/macrophages by western blot analysis (Supplementary Fig. 7b).

We further tested whether AMPK might influence macrophage skewing and liver repair. AMPKα1^−/− macrophages resulted in

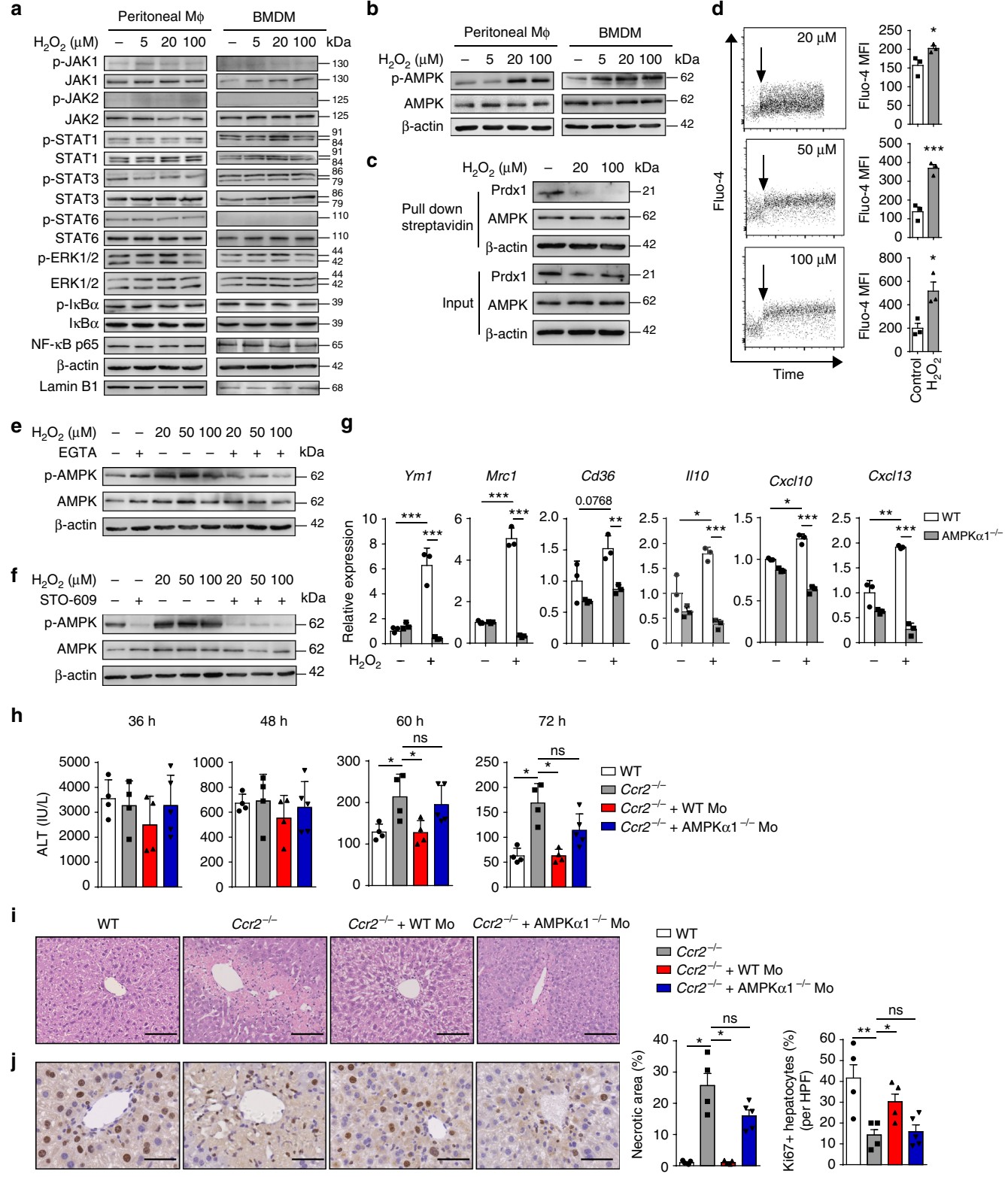

reduced expression of wound healing genes compared to WT macrophages after exposure to H₂O₂ (Fig. 6g). To examine the role of macrophagic AMPK in liver repair, we adoptively transferred WT or AMPKα⁻/⁻ monocytes to *Ccr2⁻/⁻* mice. Adoptive transfer of WT monocytes rescued the exacerbated damage and depressed hepatic regeneration in *Ccr2⁻/⁻* mice during the resolution phase, whereas transfer of AMPKα⁻/⁻

monocytes failed to do so (Fig. 6h–j). Altogether, these data suggest a link between ROS and AMPK activation in macrophage skewing and liver repair.

## Discussion

In present study, we have identified a previously unappreciated mechanism by which neutrophils orchestrate liver repair. Our

**Fig. 6** ROS promote activation of AMPK in macrophages to orchestrate liver repair. **a–c** Peritoneal macrophages or BMDMs were treated with LPS for 3 h and then were stimulated with different concentrations of $H_2O_2$ for 30 min. Immunoblot analysis of phosphorylated (p-) or total protein in lysates of macrophages and nuclear translocation of NF-κB in nuclear extracts of macrophages (**a**, **b**). Cellular proteins from peritoneal macrophages were incubated with BIAM. Protein-BIAM adducts were purified using streptavidin pull-down and then subjected to immunoblot analysis (**c**). Experiment was repeated three times. **d** $Ca^{2+}$ influx into macrophages after exposure to different concentrations of $H_2O_2$, represented by Fluo-4 intensity. $n = 3$. Experiment was repeated three times. **e**, **f** Effect of removal of extracellular $Ca^{2+}$ by EGTA (2 mM) (**e**), or inhibition of CaMKKβ by STO-609 (5 μM) (**f**) on phosphorylation of AMPK in peritoneal macrophages after 30 min $H_2O_2$ exposure. Experiment was repeated three times. **g** WT and AMPKα1$^{-/-}$ BMDMs were treated with LPS for 3 h, and then incubated with 20 μM $H_2O_2$ for 4 h. Expression of the indicated genes in $H_2O_2$-treated or untreated macrophages were determined by qPCR. $n = 3$. Experiment was repeated twice. **h**, **i** WT or AMPKα1$^{-/-}$ monocytes were adoptively transferred to APAP-challenged $Ccr2^{-/-}$ mice. Serum ALT levels at the indicated time points (**h**), histological characterization (**i**), and IHC staining for Ki67 (**j**) in liver sections at 72 h were evaluated. $n = 4, 4, 4, 5$. Experiment was repeated twice. The bar graph indicates 100 μm (**i**), 50 μm (**j**). The results represent mean ± s.e.m. (*$P < 0.05$, **$P < 0.01$, ***$P < 0.001$). $P$ values were calculated by two-tailed Student's $t$-test (**d**, **g**, **h–j**). Uncropped scans of western blots are shown in Supplementary Fig. 8

results suggest that neutrophils trigger macrophage skewing toward a reparative phenotype for optimal liver repair and the process is mediated by ROS. Our study also provides direct evidence that the cellular and molecular components involved in the initiation of inflammation also contribute to the onset of resolution and sheds light on how phagocytic populations may act together to fine-tune liver repair.

Neutrophils may engage into complex bidirectional interactions with immune cells, such as macrophages, DCs, natural killer (NK) cells, as well as with non-immune cell types, such as mesenchymal stem cells[47]. Of note, recent studies have provided new insights on the significance of neutrophil–macrophage interactions. Previous studies have shown that the coordinated interaction between neutrophils and distinct macrophage subsets are critical for antibacterial defense in a model of urinary tract infection[48]. In a mouse model of parasitic nematode infection, neutrophils were demonstrated to prime a long-lived effector macrophage phenotype, which is critical for nematode clearance[49]. Moreover, neutrophils were involved in priming macrophages for IL-1β transcription via releasing neutrophil extracellular traps (NET) in atherosclerosis[50]. Considerable advances in understanding phagocyte interaction that fuel the inflammatory cascade have been made. However, relatively little is known about the contribution of their interactions to the resolution of inflammation. Our studies provide direct evidence of neutrophil-macrophage cooperation to finely control the resolution of inflammation. Further studies will clarify their functional significance in various contexts.

Previous studies have shown that neutrophils and distinct monocyte/macrophage subsets accumulated in the liver after APAP-induced injury[16,51]. The diverse roles of two different infiltrated monocyte/macrophage populations have been extensively studied[52–56]. Ly6C$^{hi}$CX$_3$CR1$^{lo}$ monocytes/macrophages exhibited a pro-inflammatory phenotype and mediated the development of liver injury, while Ly6C$^{lo}$CX$_3$CR1$^{hi}$ macrophages dampened inflammation and promoted tissue repair. However, the in vivo function of neutrophils has been less well characterized, compared to that of macrophages. Increasing amounts of data indicate that neutrophils do not contribute to the liver injury during the early phase after APAP overdose[57]. Nevertheless, the role of neutrophils during the resolution phase remains largely unexplained. Here, we provide evidence that neutrophils are required for optimal liver regeneration and repair through promoting macrophage conversion. The time lag between neutrophil accumulation and macrophage conversion may be due to the fact that it takes time for macrophages to alter their phenotypes in response to local tissue signals. Several studies have shown that macrophages require time for their phenotypic conversion. For example, in a mouse model of peritoneal inflammation, it took 8 weeks for recruited CX$_3$CR1$^+$ monocyte-derived cells to adopt a

tissue-resident peritoneal macrophage phenotype[30]. In a thermal-induced liver injury, CCR2$^{hi}$CX$_3$CR1$^{lo}$ monocytes were recruited to the injured area at 24 h and transitioned into CCR2$^{lo}$CX$_3$CR1$^{hi}$ monocytes at 48 h[13].

ROS, traditionally viewed as harmful to cells, are now appreciated to have pleiotropic biological effects on various cellular processes, including limiting inflammation and autoimmune responses[58,59]. A mutation in one of the subunits of the Nox2 complex resulted in increased susceptibility to severe arthritis, suggesting that ROS are beneficial to control self-reactive T cells[60]. Furthermore, ROS have been reported to be involved in axonal regeneration and Xenopus tadpole tail regeneration[61,62]. Yet our data suggest a more expansive role: Because genetic deletion of Nox2 impaired the resolution of hepatic damage and delayed liver regeneration, we propose that ROS may play important roles in modulating the resolution of inflammation. A previous study showed that neutrophil activation may be critical for liver repair following APAP overdose[18], which support our findings. However, the study did not show significant difference in serum ALT levels between WT and Nox2$^{-/-}$ mice during the resolution phase. The divergent conclusions may be related to different progression of injury. Compared to the previous study, we used more WT and Nox2$^{-/-}$ mice to assess their difference in liver recovery and observed over a longer period of time until serum ALT concentrations returned to normal levels.

Macrophages adopt their distinct functional phenotypes in response to micro-environmental cues. Alternatively activated or anti-inflammatory macrophages are known to be induced by Th2 cell cytokines, typically IL-4 and IL-13, which can be produced by Th2 cells, eosinophils, basophils, mast cells, natural killer T (NKT) cells, and innate lymphoid cells[63]. In adipose tissue, eosinophils contribute to the generation of alternatively activated macrophages in an IL-4- and IL-13-dependent manner[64]. In allergic skin, basophil-derived IL-4 are involved in the conversion from inflammatory monocytes to anti-inflammatory macrophages[65]. In present study, we demonstrate that in the setting of APAP-induced liver injury, neutrophils mediate macrophage skewing toward a reparative phenotype. Previous studies have shown similar roles of neutrophils in regulation of macrophage phenotype[66], even though no direct evidence for this in vivo was provided. Thus, in addition to those well-known cell types reported previously, neutrophils can contribute to phenotypic conversion of macrophages and therefore orchestrate tissue repair. On the other hand, we observed that ROS expressed predominantly by neutrophils could be a novel local signal to dictate macrophage phenotype. Although several studies have reported that ROS could be intrinsic cell signals that regulate macrophage differentiation and polarization[67], we showed that extrinsic ROS, which are mainly released by neutrophils, triggered

macrophage skewing toward a reparative phenotype. However, we cannot rule out the possibility that additional factors other than ROS might contribute to the polarizing effects on macrophages.

Mechanically, we found that ROS promoted macrophage skewing toward a reparative phenotype, possibly by activating AMPK. AMPK deficiency in macrophages prevented the acquisition of a pro-resolving phenotype upon upon $H_2O_2$ treatment. We observed that $H_2O_2$ induced $Ca^{2+}$ influx into macrophages and CaMKKβ inhibition resulted in reduced activation of AMPK, which suggest that $Ca^{2+}$-CaMKKβ may be responsible for ROS-mediated AMPK activation. It remains to be definitely demonstrated how ROS regulate $Ca^{2+}$ influx and how ROS-induced AMPK activation mediates macrophage reprogramming. Previous studies have demonstrated that $H_2O_2$ evoked $Ca^{2+}$ influx and mediated amplification of Erk activation and NF-κB nuclear translocation in human U937 monocytes, leading to chemokine production[68]. In our hands, $H_2O_2$ treatment did not enhance activation of Erk or NF-κB (Fig. 6a). This may be a result of concentration-dependent effects: high concentrations (250 μM) of ROS in previous studies induced significant activation of NF-κB, whereas low or moderate concentrations of ROS may exert regulatory effect by a different mechanism. Since it has been observed that different levels of $H_2O_2$ can induce distinct responses within a cell[43], we may speculate that a certain level of ROS is required for promoting reparative macrophage polarization. Furthermore, AMPK activation can modulate many essential metabolic pathways through regulating a number of transcriptional factors and cofactors, such as PGC-1α, CREB, SIRT1, and PPARs[69]. Since metabolic reprogramming has been reported to orchestrate macrophage polarization and macrophage functional plasticity[70], it is tempting to speculate that ROS-induced AMPK activation may mediate the phenotypic transition of macrophages by altering their metabolic states.

## Methods

**Mice**. C57BL/6 wild-type (WT) mice were purchased from Charles River in Beijing (Vital River). $Ccr2^{-/-}$ (B6.129S4-$Ccr2^{tm1Ifc}$/J) mice, $Nox2^{-/-}$ (B6.129S-$Cybb^{tm1Din}$/J or gp91$^{phox-/-}$) mice and $Rosa26^{tdTomato}$ mice were purchased from Jackson Laboratory. $Gcsf^{-/-}$ ($Csf3^{tm1Ard}$) mice were kindly provided by Dr. Ian Wicks (Ludwig Institute for Cancer Research Ltd., The Walter and Eliza Hall Institute of Medical Research). CD45.1/Ly5.1 mice were generously provided by Dr. Mingzhao Zhu (Institute of Biophysics, Chinese Academy of Sciences). $Cx3cr1^{CreERT2-IRES-EYFP}$ mice were kindly provided by Dr. Wenbiao Gan (New York University Medical Center). AMPKα1$^{-/-}$ mice were provided by Collaborative Innovation Center of Model Animal Wuhan University. All the mouse strains used were on a C57BL/6 background. Male mice and their littermate control between six- to eight-week-old were maintained in a specific pathogen-free condition. All experimental procedures in mice were approved by the Institutional Animal Care and Utilization Committee-approved protocols (IACUC-2015-NCPSB001) and complied with all relevant ethical regulations.

For neutrophil depletion, mice were injected i.p. with 200 μg anti-Ly6G (clone 1A8, BD Biosciences) 6 h after APAP injection.

For labeling $Cx3cr1^{Cre+}$ cells, $Cx3cr1^{CreERT2-EYFP/+}Rosa26^{tdTomato/+}$ mice were given a single dose of 10 mg tamoxifen (Sigma-Aldrich) via oral gavage 1 day before APAP challenge.

For in vivo ROS neutralization in $Cx3cr1^{CreERT2-EYFP/+}Rosa26^{tdTomato/+}$ mice, mice were injected intraperitoneally (i.p.) with two doses of NAC (100 mg kg$^{-1}$, Sigma-Aldrich) at the time of APAP challenge and 36 h after APAP challenge.

**APAP-induced hepatotoxicity and Assays for liver injury**. Acetaminophen (Sigma-Aldrich) solution was made fresh for each experiment. In brief, mice were fasted 15–16 h and injected i.p. with acetaminophen at 400 mg kg$^{-1}$. For assay of serum liver enzyme ALT, mice were anesthetized and bled from the retroorbital venous plexus at the indicated time points. Serum ALT levels were evaluated by diagnostic kits. Total GSH and catalase in whole liver homogenates were measured using a Glutathione and Catalase Assay Kit, respectively.

**Generation of mixed bone marrow chimeras**. WT recipient mice were irradiated twice with 5 Gy and reconstituted i.v. with 50% WT + 50% $Gcsf^{-/-}$, or 50%

$Nox2^{-/-}$ + 50% $Gcsf^{-/-}$ bone marrow cells (total transplanted cells were 10$^7$ cells per mouse). Recipients were left for 8 weeks before being used in experiments.

**Adoptive transfer of monocytes or neutrophils**. To adoptively transfer Ly6C$^{hi}$ monocytes, viable CD45.1$^+$CD115$^+$CD11b$^+$Ly6C$^{hi}$ monocytes from bone marrow were sorted by FACS (Fluorescence-activated cell sorting, using BD FACSAria III). $2 \times 10^6$ monocytes were adoptively transferred into CD45.2 recipient mice by intravenous injection 4 h after APAP injection.

To adoptively transfer neutrophils, neutrophils from bone marrow were purified using the Neutrophil Isolation Kit (Miltenyi Biotec, #130-097-658). $8 \times 10^6$ neutrophils were adoptively transferred into APAP-challenged recipient mice by intravenous injection.

**Primary cell isolation**. Mice were anesthetized, and livers were perfused in situ via the portal vein with perfusion buffer (HBSS), followed by a digestion buffer (HBSS supplemented with 0.05% collagenase, 2.15 mM CaCl$_2$, 4 mM MgCl$_2$, 0.028% DNase I). After digestion, the livers were homogenized and filtered through a 70 μm nylon mesh. The cell suspension was centrifuged at 500×$g$ for 5 min, the resulting cell pellet from one liver was re-suspended in 15 ml 35% Percoll (GE Healthcare) containing 100 U ml$^{-1}$ heparin. The cell suspension was centrifuged at 500×$g$ for 15 min, and the cell pellet containing leukocytes was harvested and re-suspended in 1 ml red blood cell lysis solution (BD Biosciences). After incubation for 3 min, cells were washed twice in RPMI 1640 containing 3% fetal bovine serum (FBS).

Primary mouse hepatocytes were isolated by 2-step collagenase perfusion method as described above. After perfusion with digestion buffer, the liver was dissociated in suspension buffer and filtered through a 70 μm nylon mesh. Hepatocytes were collected by centrifugation at 50×$g$ for 3 min.

Bone marrow cells were prepared and cultured in DMEM supplemented with 10% FBS and M-CSF (50 ng ml$^{-1}$, Peprotech) for macrophage differentiation for 7 days. In other cases, thioglycollate (Sigma-Aldrich) elicited peritoneal macrophages were isolated.

**Flow cytometry and FACS sorting**. For flow cytometric analysis, hepatic cells were first incubated with Fcγ receptor blocker (CD16/32, eBioscience), then stained with specific mAbs at indicated concentration. The following antibodies were used: fluorescently-conjugated antibodies directed against mouse (all from eBioscience unless specified otherwise): CD45 (30-F11, dilution 1/100), Ly6G (1A8, dilution 1/100, BD Biosciences), F4/80 (BM8, dilution 1/80), CD11b (M1/70, dilution 1/200), CD3 (17A2, dilution 1/100), CD19 (eBio1D3, dilution 1/100), CD49b (DX5, dilution 1/100), γδTCR (eBioGL3, dilution 1/100), CD11c (N418, dilution 1/100), Siglec-F (E50-2440, dilution 1/200, BD Biosciences), Ly6C (HK1.4, dilution 1/200), CD64 (X54-5/7.1, dilution 1/200), MerTK (DS5MMER, dilution 1/50), CX$_3$CR1 (SA011F11, dilution 1/200, Biolegend), Gr-1(RB6-8C5, dilution 1/200), MHC-II (M5/114.15.2, dilution 1/100), CD45.1 (A20, dilution 1/40), CD45.2 (104, dilution 1/40) and their corresponding isotype controls. Cell populations were defined as: neutrophils (PMNs) were identified as CD11b$^+$Ly6C$^{int}$Ly6G$^+$. Kupffer cells (KCs) were identified as F4/80$^{hi}$CD11b$^{lo}$. Monocyte-derived macrophages were identified as F4/80$^{lo}$CD11b$^{hi}$. T cells were identified as CD3$^+$. B cells were identified as CD19$^+$. NKT cells were identified as CD3$^+$DX5$^+$. NK cells were identified as CD3$^-$DX5$^+$. γδT cells were identified as CD3$^+$γδTCR$^+$. DC cells were identified as CD11c$^+$MHC-II$^+$. Eosinophils were identified as CD11b$^+$F4/80$^{int}$Siglec-F$^+$. The stained cells were analyzed on an LSRFortessa cell analyzer (BD Biosciences). The acquired data were analyzed with FlowJo software (Tree Star).

For cell purification, hepatic leukocytes were isolated as described above, followed by staining for cell-surface markers. Then, hepatic neutrophils and macrophage subpopulations were sorted by FACSAria III (BD Biosciences). FACS sorting routinely yielded cell purity levels of over 95%.

**Determination of hepatocyte proliferation**. Primary mouse hepatocytes were isolated from WT mice. Neutrophils and distinct macrophage subsets were isolated at the indicated time points. For co-culture studies, isolated hepatocytes were plated at a density of $1 \times 10^5$ cells ml$^{-1}$, seeded with $2 \times 10^6$ cells ml$^{-1}$ macrophages. Culture conditions consisted of RPMI1640 (Invitrogen) supplemented with 10% FBS, streptomycin (100 U ml$^{-1}$) and penicillin (100 U ml$^{-1}$). Conditioned medium (CM) was collected from 200,000 neutrophils or distinct macrophage subsets, filtered through a 0.22 μm filter, and added to 10,000 hepatocytes. Hepatocytes were treated with neutrophil or macrophage-derived CM for 12 h, followed by EdU (5-ethynyl-2′-deoxyuridine, 20 μM) pulsing for an additional 36 h. Hepatocytes undergoing DNA synthesis were visualized by the EdU Imaging Kit (Life Technologies). Imaging was performed using Olympus IX71 inverted fluorescence microscopes and EdU-positive cells were quantified by ImageJ software.

**Neutrophil/macrophage co-culture**. To assess the effect of hepatic neutrophils on macrophage phenotype, Ly6C$^{hi}$CX$_3$CR1$^{lo}$ monocytes/macrophages were stimulated with neutrophil-derived CM for 6 h, subsequently, Ly6C$^{hi}$CX$_3$CR1$^{lo}$ monocytes/macrophages were washed with PBS and RNA was extracted. Gene expression was detected by qPCR. In a separated experiment, hepatic neutrophils were cultured with hydrogen peroxide degrading enzyme catalase (5 μM, Sigma

Aldrich) and the CM were treated with Ly6C$^{hi}$CX$_3$CR1$^{lo}$ monocytes/macrophages for 6 h.

In some experiments, Ly6C$^{hi}$CX$_3$CR1$^{lo}$ monocytes/macrophages were treated with CM from WT or Nox2$^{-/-}$ neutrophils for 12 h, subsequently, the supernatants were added to hepatocytes for 48 h to determine the proliferation of hepatocytes. Hepatocyte proliferation induced by the supernatants from Ly6C$^{hi}$CX$_3$CR1$^{lo}$ monocytes/macrophages that had been co-cultured with neutrophils compared to supernatants from Ly6C$^{hi}$CX$_3$CR1$^{lo}$ monocytes/macrophages or neutrophils cultured alone.

**ROS and Ca$^{2+}$ influx detection**. Intracellular ROS were detected by CM-H$_2$DCFDA (Life technologies, C6827). Liver non-parenchymal cells were stained with 5 μM H$_2$DCFDA in 37 °C incubator for 30 min followed by washing twice in PBS before FACS analysis. The concentration of Hydrogen peroxide was measured by an Amplex Red Hydrogen Peroxide/Peroxidase Assay Kit (Life Technologies, A22188).

Calcium influx was determined using a Fluo-4 Direct Calcium Assay Kit (Thermo Fisher, F10471). Fluo-4 was recorded over time on a BD FACSVerse flow cytometer.

**Immunohistochemistry**. Liver specimens from mice exposed to various treatments were fixed in 4% paraformaldehyde and embedded with paraffin for further analysis. The sections were stained with hematoxylin and eosin or with monoclonal Rabbit anti-mouse Ki67 (clone SP6, ab16667, dilution 1/200, Abcam). Images of liver slides were obtained on a Nano Zoomer Slide Scanner. Necrotic areas and percentage of Ki67 positive cells were quantified by ImageJ software.

**Immunofluorescence**. Mouse liver tissues were fixed in 4% paraformaldehyde overnight at 4 °C and then embedded in OCT (Sakura). Five micrometer Frozen sections were prepared using a Cryotome FSE cryostat (Thermo-Fisher Scientific). The tissue sections were incubated in the blocking buffer (5% donkey serum, 0.3% Triton-X 100 in PBS) at room temperature for 1 h followed by the staining with primary antibodies. The following primary antibodies were used: Rat anti-mouse F4/80 (clone CI:A3-1, ab6640, dilution 1/500, Abcam) and Rabbit anti-mouse phosphor-AMPKα (Thr172) antibody (clone 40H9, #2535, dilution 1/200, Cell Signaling Technology). Then slides were washed and incubated for 1 h with the following secondary antibodies: donkey anti-rat Alexa Fluor 488 and donkey anti-rabbit Alexa Fluor 594 (Jackson ImmunoResearch Laboratories). Sections were counterstained with 4′,6-diamidino-2-phenylindole dihydrochloride (DAPI) before being mounted. All immunofluorescence staining was performed in the dark. Imaging was performed using a Zeiss LSM 880 and images were processed using Zeiss ZEN software.

**Quantitative RT-PCR**. Total RNA was extracted using RNeay Kit (Qiagen) according to the manufacturer's protocol. Typically, 1 μg of total RNA was reverse transcribed into cDNA using Reverse Transcription kit (Promega). The cDNA was used for quantitative qPCR analysis on an iCycler iQ5 Real-Time PCR detection system (BioRad) as manufacturer's instructions. The expression of target gene was normalized to the expression of the housekeeping gene, *Gapdh*. Relative gene expression was calculated using the standard 2$^{-\Delta\Delta Ct}$ method. Primer sequences are shown in Supplementary Table 1.

**Western blot analysis and label of free cysteine thiols**. Macrophage protein extracts were prepared according to standard protocols. Cell lysates were separated by 10% SDS-PAGE and transferred to polyvinylidene difluoride membranes (Millipore). The following antibodies (all from Cell Signaling Technology unless specified otherwise) were used: p-JAK1 (no. 74129, dilution 1/200), JAK1 (no. 3344, dilution 1/200), p-JAK2 (no. 8082, dilution 1/200), JAK2 (no. 3230, dilution 1/200), p-STAT1 (no. 7649, dilution 1/1000), STAT1 (no. 9172, dilution 1/1000), p-STAT3 (no. 9134, dilution 1/1000), STAT3 (no. 4904, dilution 1/1000), p-STAT6 (no. 56554, dilution 1/500), STAT6 (no. 5397, dilution 1/500), p-ERK1/2 (no. 9101, dilution 1/1000), ERK1/2 (no. 3230, dilution 1/1000), p-IκBα (no. 2859, dilution 1/1000), IκBα (no. 4814, dilution 1/1000), NF-κB p65 (no. 8242, dilution 1/1000), Lamin B1 (no. 13435, dilution 1/1000), p-AMPKα (no. 2535, dilution 1/500), AMPKα (no. 2532, dilution 1/500), Prdx1 (Abcam, ab15571, dilution 1/1000) and β-actin (Sigma Aldrich, A5441, dilution 1/3000).

Nuclear translocation of NF-κB was determined by Western blot with NF-κB p65-specific antibody. Nuclear extracts were prepared using NE-PER nuclear and cytoplasmic extraction reagents (Thermo Scientific).

The extent of free (unoxidized) cysteine residues within AMPK was determined by the BIAM labeling assay[44]. Cell lysates were incubated with BIAM (200 μM) for 1 h at room temperature, and then BIAM-protein conjugates were precipitated with streptavidin-agarose overnight at 4 °C. BIAM-protein adducts were extracted from streptavidin-agarose by boiling for 10 min and then subjected to Western blot analysis. Uncropped scan of western blots are shown in Supplementary Fig. 8.

**Statistical analysis**. Statistical analysis was performed with GraphPad Prism v5 software. Data are presented as mean ± s.e.m. The number of samples for each experiment and the replicate number of experiments are reported in the figure legends. Statistical significance was calculated by unpaired, two-tailed, Student's *t*-test or one-way analysis of variance (ANOVA) where appropriate. A *P* value < 0.05 was considered statistically significant. No formal randomization was used and animals were unbiasedly assigned into different treatment groups. Group allocation and outcome assessment was performed in a blinded manner. No exclusion criteria were applied, and all samples were included in data analysis.

**Reporting summary**. Further information on experimental design is available in the Nature Research Reporting Summary linked to this article.

## Data availability
All the relevant data supporting the findings of this study are available within the article, or from the corresponding author on reasonable request.

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

## Acknowledgements

This work was supported by National Key R&D Program of China (2018YFA0507500), Chinese State Key Program in Basic Research (2014CBA02000, 2013CB910802), Chinese National Natural Science Foundation Projects (81500481, 81400615) and Beijing Science Program for the Top Young (2015000021223TD04). We thank Dr. Ian Wicks (Walter and Eliza Hall Institute of Medical Research) and Dr. Ashley Dunn (Ludwig Institute for Cancer Research Ltd.) for Gcsf−/− mice, Dr. Mingzhao Zhu (Institute of Biophysics, Chinese Academy of Sciences) for CD45.1 mice, and Dr. Wenbiao Gan (New York University Medical Center) for *Cx3cr1*<sup>CreERT2-IRES-EYFP</sup> mice. We thank Collaborative Innovation Center of Model Animal Wuhan University for providing AMPKα1−/− mice. We thank Flow Cytometry Facility, Animal Facility (Mr. Chen Qiu) and Imaging Facility (Ms. Ping Wu) of National Center for Protein Sciences. Beijing (NCPSB) for their assistance.

## Author contributions

W.Y. performed the experiments, analyzed the data and wrote the manuscript. Y.T. performed the experiments and analyzed the data. Y.W., X.Z., W.Y., D.Z., L.F., C.T. and J.Y. provided technical or material support. L.T. and F.H. conceived and supervised the study.

## Additional information

**Competing interests:** The authors declare no competing interests.

