## [Peer Review File · Nature Communications]

Reviewers' comments:

Reviewer #1 (APAP mouse model, macrophage, ROS)(Remarks to the Author):

Overall, this manuscript is not cohesive and, with over 100 figures and panels, is very difficult to follow. Moreover, data does not always support the conclusions. The work addresses two issues: 1) neutrophils participate in repair pathways in the liver following acetaminophen intoxication, possibly by regulating phenotypic switching of M1 and M2 macrophages, and 2) reactive oxygen species (ROS) regulate this process. The literature would better be served if the authors divided the manuscript and greatly reduced the number of Figures/panels. The issue of neutrophils regulating phenotypic switching is of interest, although the authors have not addressed significant literature on this topic. The issue of neutrophil-derived ROS as regulators of macrophage function is poorly developed in this manuscript; significantly more work needs to be done to convince the reader that neutrophil ROS are responsible for mediating liver repair via macrophages. The data does not support the idea that "Neutrophils license macrophages for orchestrating liver repair by release reactive oxygen species".

In general, the authors did not adequately address comments of Reviewer #1. For example, the reviewer correctly pointed out that although N-acetyl-L- cysteine (NAC) is an inhibitor of ROS, it also acts as an antidote for APAP overdose through a number of potential mechanisms other than inhibition of ROS. Importantly, it is now known that NAC is active and used clinically because it directly detoxifies acetaminophen (by binding the acetaminophen metabolite NAPQI). NAC cannot be used to simply detoxify ROS, the authors results with NAC are due to acetaminophen detoxication. Note also that given that NOX deficient mice (mice deficient in NADPH oxidase) inhibit the ability of neutrophils, as well as macrophages, from producing ROS, the authors cannot conclude that changes in toxicity of acetaminophen in these knockout mice are due to ROS production from neutrophils.

Additional points: Supplement, Fig 1, panels A and B: the x-axis in these panels is misplotted and should be on a linear time scale. Fig 3, panel b, catalase controls are missing, what is being measured in liver homogenates as they contain hydrogen peroxide generating material such as mitochondria and hydrogen peroxide degrading enzymes such as catalase. Fig. 5, without appropriate controls (such as the addition of catalase) one cannot conclude that neutrophil derived ROS mediates the conversion of pro-inflammatory macrophage ex vivo; these are critical controls in experiment using conditioned medium that were not presented. Fig. 6, no scale is presented for panel d, data does not appear to be coordinate with the right panel in the Figure. It is not helpful to compare the effects of ROS using LPS treated peritoneal macrophages or BMDM (Fig. 6) when the entire paper is on liver macrophages and acetaminophen, they are not the same macrophage populations and do not respond as liver macrophages. Fig. 6, panel a, appears to be present almost all negative data; activation of many signaling pathways occur on different time scales, time courses would need to be done to conclude that "Activation of [various kinases] in H₂O₂ treated macrophages was similar to that in untreated macrophages" (page 14). Does acetaminophen/neutrophils/hydrogen peroxide activate AMPK in isolated hepatic macrophage populations (confocal images shown in Fig. 6, panel h, are not convincing and not comparable with isolated peritoneal and BMDM macrophages). Supplementary Fig. 2, no data is presented for $p < 0.01$. Supplement, Fig 1 appears to show significant changes in immune cell populations in addition to macrophages, thus the basis for the author's statement that "other immune cell populations did not change significantly" (page 5) is not clear. Fig. 1, neutrophils may contribute to tissue repair, but there is no evidence that "neutrophil depletion delays...liver repair after acute injury" (legend to Fig. 1). In chimera studies using Gcsf and Nox2 ko bone marrow, there was no consideration of the fact that there is partial depletion of macrophages and this can markedly alter

the conclusions from these studies.

Reviewer #2 (ROS, neutrophil)(Remarks to the Author):

The authors addressed all of the comments of the reviewers and significantly improved the manuscript.

Reviewer #3 (Neutrophil biology, liver inflammation)(Remarks to the Author):

This is an interesting manuscript with an interesting relationship between neutrophils and monocytes/macrophage in sterile injury and repair. I have a number of general comments that should be addressed.

1) Not sure that a Ly6C high macrophage is an accepted term. Generally, these are referred to as monocytes. Throughout the manuscript the authors refer to Ly6C high and Ly6C low macrophage. However, immunologists would see this as markers of monocytes and macrophage respectively. While this could just be terminology it could also be a completely alternative interpretation i.e., neutrophils help monocytes mature into macrophage. The authors must demonstrate at the beginning of their results that they truly are looking at macrophage and macrophage i.e., is F4/80 and/or MerTK and/or other macrophage markers on the surface of both Ly6C high and Ly6C low cells.

2) In the abstract and throughout the manuscript the authors make the following claim verbatim: "We mechanistically demonstrate that neutrophil-derived reactive oxygen species (ROS) trigger the phenotypic switch between macrophages via AMP-activated protein kinase (AMPK)" This is not shown in this manuscript. The authors show that AMPK is oxidized/activated but whether it is responsible in the switch is not examined. The authors only make one sentence statement saying that the knockout has fewer repair genes. This needs to be better fleshed out. Was there less repair in this knockouts?

3) I am not sure the results of depletion of neutrophils in CCR2^{-/-} mice provide a clear conclusion that neutrophil repair is related to monocytes.

Point-by-point Response to Reviewers

Reviewer comments:

Reviewer #1 (APAP mouse model, macrophage, ROS)(Remarks to the Author):

Question 1:

Overall, this manuscript is not cohesive and, with over 100 figures and panels, is very difficult to follow. Moreover, data does not always support the conclusions. The work addresses two issues: 1) neutrophils participate in repair pathways in the liver following acetaminophen intoxication, possibly by regulating phenotypic switching of M1 and M2 macrophages, and 2) reactive oxygen species (ROS) regulate this process. The literature would better be served if the authors divided the manuscript and greatly reduced the number of Figures/panels. The issue of neutrophils regulating phenotypic switching is of interest, although the authors have not addressed significant literature on this topic. The issue of neutrophil-derived ROS as regulators of macrophage function is poorly developed in this manuscript; significantly more work needs to be done to convince the reader that neutrophil ROS are responsible for mediating liver repair via macrophages. The data does not support the idea that “Neutrophils license macrophages for orchestrating liver repair by release reactive oxygen species”.

Response:

First, we wish to express our sincere thanks for your constructive comments and suggestions. Your questions and suggestions have been thoroughly discussed among all authors and we addressed your concerns in the following.

We have re-organized the manuscript and greatly reduced redundant data to better present the main information in a streamlined logical flow, as you suggested.

Regarding the issue of neutrophils regulating macrophage phenotypic switching, we realized that we did not elaborate on the importance of this main finding. We have discussed the topic in the Discussion Section in the revised manuscript.

We thank you for pointing out the lack of strong evidence for the role of neutrophil-

derived ROS. As far as we know, there are two classical approaches to assess the role of specific types of cell-derived mediators under in vivo conditions. The ideal is to create neutrophil-specific deletions using Cre-lox system. The other is to generate bone marrow chimeras by transplanting mixed bone marrow cells from a neutrophil-deficient and a gene knockout mouse into previously lethally irradiated wild type mice. In our previous study, we had considered the use of lineage-specific conditional knockout mice to assess the pro-repair role of neutrophil-derived ROS. We found that three transgenic mice may be used (e.g. MRP8 (S100A8)-Cre^{1, 2, 3}, GE (Ela)-Cre^{4, 5} and LysM-Cre^{6, 7}) for conditional deletion of target genes in neutrophils. However, it is important to note that high specificity cannot always be obtained when using these neutrophil-specific conditional knockout mice⁸. MRP8-Cre mice have been reported to allow for Cre-mediated deletion of loxP-flanked target genes in granulocytes, monocytes and a fraction of granulocyte-macrophage progenitor^{1,2,9}. GE-Cre mice can be used for conditional gene targeting in the myeloid lineage, including granulocytes and inflammatory macrophages^{4, 10}. LysM-Cre mice have been widely used to genetically target myeloid cells, including granulocytes, macrophages, inflammatory monocytes and myeloid-derived dendritic cells^{7, 11}. Given the limited specificity of these lineage-specific conditional knockout mice, they may not be suitable for us to assess the role of neutrophil-derived ROS in liver repair.

Generation of mixed bone marrow chimera provides an alternative method to investigate the importance of a molecule in certain cell types⁸. This approach is also widely used in many studies^{12, 13, 14}. Thus, we chose bone marrow chimeras instead of neutrophil-specific conditional knockout mice to examine the role of neutrophil-derived ROS in liver repair. We found that *Nox2*^{-/-}/*Gcsf*^{-/-} mixed bone marrow chimeric mice profoundly impaired liver regeneration and macrophage conversion during the resolution phase, as compared to WT/*Gcsf*^{-/-} chimeras (**Fig. 4a-e**).

Considering that there are partial depletion of Nox2 in bone marrow cells other than neutrophils in chimera models, as you mentioned in **Question 13**, we have performed an adoptive transfer experiment to confirm the role of neutrophil-derived ROS. We adoptively transferred WT or *Nox2*^{-/-} neutrophils into neutrophil-depleted mice (**Fig. 4f**,

Supplementary Fig.4f, g). Adoptive transfer of WT neutrophils rescued the exacerbated damage and depressed hepatic regeneration in neutrophil-depleted mice during the resolution phase (**Fig. 4g-i**). In contrast, transfer of *Nox2*^{-/-} neutrophils failed to do so. Moreover, transfer of WT but not *Nox2*-deficient neutrophils promoted macrophage conversion in neutrophil-depleted mice (**Fig. 4j**). These results may suggest that ROS-producing neutrophils contribute to macrophage skewing and liver repair. However, considering that we cannot use the ideal lineage-specific deletion method to obtain more unambiguous evidence about “neutrophil-derived ROS”, we have tuned down our statements of relevant conclusions in the revised manuscript.

Question 2:

In general, the authors did not adequately address comments of Reviewer #1. For example, the reviewer correctly pointed out that although N-acetyl-L- cysteine (NAC) is an inhibitor of ROS, it also acts as an antidote for APAP overdose through a number of potential mechanisms other than inhibition of ROS. Importantly, it is now known that NAC is active and used clinically because it directly detoxifies acetaminophen (by binding the acetaminophen metabolite NAPQI). NAC cannot be used to simply detoxify ROS, the authors results with NAC are due to acetaminophen detoxication. Note also that given that NOX deficient mice (mice deficient in NADPH oxidase) inhibit the ability of neutrophils, as well as macrophages, from producing ROS, the authors cannot conclude that changes in toxicity of acetaminophen in these knockout mice are due to ROS production from neutrophils.

Response:

Thanks for your valuable comments. We acknowledge that NAC is not only an inhibitor of ROS, but also an antidote for APAP overdose. Thus, we agree that the use of NAC to assess the role of ROS in liver repair is not appropriate. However, it is worth noting that the increased severity of APAP-induced liver injury during the resolution phase after NAC treatment could not be simply explained by APAP detoxification. If NAC detoxified APAP, liver damage would be mitigated rather than aggravated during the resolution phase. Even so, using NAC could not clearly demonstrate the role of ROS in

liver repair. Thus, we removed all the results of NAC related to liver repair.

On the other hand, we used a more appropriate strategy to ascertain the importance of ROS. *In vivo*, we took advantage of *Nox2*^{-/-} mice, in which the gp91 subunit of NADPH oxidase is deleted (**Fig. 3d-g**). *In vitro*, we used catalase to degrade hydrogen peroxide as well as *Nox2*^{-/-} neutrophils for neutrophil-macrophage co-culture experiments (**Fig. 5a, b**).

Regarding the issue of neutrophil-derived ROS, we have performed adoptive transfer experiments in order to provide more convincing evidence, as mentioned in Response to **Question 1**.

Additional points:

Question 3:

Supplement, Fig 1, panels A and B: the x-axis in these panels is misplotted and should be on a linear time scale.

Response:

We're sorry for the mistakes and we have corrected the error as below.

Question 4:

Fig 3, panel b, catalase controls are missing, what is being measured in liver homogenates as they contain hydrogen peroxide generating material such as mitochondria and hydrogen peroxide degrading enzymes such as catalase.

Response:

Thanks for your good suggestions. We have measured the catalase concentration in liver homogenates and found that there was no significant difference in hepatic catalase concentrations between normal and APAP-challenged mice (**Supplementary Fig.4c**).

Question 5:

Fig. 5, without appropriate controls (such as the addition of catalase) one cannot conclude that neutrophil derived ROS mediates the conversion of pro-inflammatory macrophage ex vivo; these are critical controls in experiment using conditioned medium that were not presented.

Response:

Thanks for your good suggestions. We have added catalase to degrade hydrogen peroxide in the neutrophil-macrophage co-culture system. Neutrophil-conditioned medium promoted pro-inflammatory Ly6C^{hi}CX3CR1^{lo} monocytes/macrophages skewing toward a reparative phenotype, but the addition of catalase induced a decrease of pro-resolving marker and an increase of pro-inflammatory marker expression (**Fig. 5a**).

Question 6:

Fig. 6, no scale is presented for panel d, data does not appear to be coordinate with the right panel in the Figure.

Response:

Thanks for your good suggestions. We're sorry for the mistakes. Mean intensity of Ca²⁺ influx after various concentrations of H₂O₂ stimulation has been presented in the revised manuscript (**Fig. 6d**).

Question 7:

It is not helpful to compare the effects of ROS using LPS treated peritoneal macrophages or BMDM (Fig. 6) when the entire paper is on liver macrophages and acetaminophen, they are not the same macrophage populations and do not respond as liver macrophages.

Response:

Thanks for your comments. We agree that peritoneal macrophages and BMDMs cannot completely mimic liver macrophages. In the previous submission process, some

reviewers suggested that we should explore the molecular mechanisms by which ROS regulate macrophage skewing in vitro. Because the number of hepatic recruited macrophages is too small for analyzing protein expression with various stimulating concentrations or at different time points, and cell viability after FACS sorting may be unstable in each experiment, it is very difficult to study molecular mechanisms using these macrophage populations isolated from livers. For these reasons, we used primary peritoneal macrophages and BMDMs in vitro mechanistic studies. Indeed, several published papers have used these cells instead of macrophage populations isolated from tissues for in vitro studies^{15, 16, 17, 18}. Considering primary peritoneal macrophages and BMDMs are not the same macrophage populations isolated from livers, our results only provide a hint about the molecular link between ROS and AMPK in macrophage skewing and we have tuned down our statements of relevant conclusions in the revised manuscript.

Question 8:

Fig. 6, panel a, appears to be present almost all negative data; activation of many signaling pathways occur on different time scales, time courses would need to be done to conclude that “Activation of [various kinases] in H₂O₂ treated macrophages was similar to that in untreated macrophages” (page 14).

Response:

Thanks for your good suggestions. We have measured protein level changes of various kinases in response to H₂O₂ at the indicated time points (**Below Figures a, b**). We found that at the time points observed, 20 μM H₂O₂ stimulation induced activation of AMPK within 15-120 min, but H₂O₂ did not significantly affect the phosphorylation of JAK1, JAK2, STAT3, STAT6, ERK1/2 and NF-κB. Additionally, 20 μM H₂O₂ stimulation may induce phosphorylation of STAT1 in a short time (within 15 min). Your concerns remind us that the conclusion is not accurate in the previous manuscript, so we changed the statement to “H₂O₂ treatment at indicated concentrations had no significant effects on the activation of JAK1, JAK2, STAT1, STAT3, STAT6, ERK1/2 and NF-κB at the time points observed”.

Question 9:

Does acetaminophen/neutrophils/hydrogen peroxide activate AMPK in isolated hepatic macrophage populations (confocal images shown in Fig. 6, panel h, are not convincing and not comparable with isolated peritoneal and BMDM macrophages).

Response:

Thanks for your good suggestions. We performed western blots of p-AMPK in isolated hepatic macrophage populations. Consistent with *in vivo* labelling, increased phosphorylation of AMPK was observed in 72 hr Ly6C^{lo}CX₃CR1^{hi} macrophages compared to 24 hr Ly6C^{hi}CX₃CR1^{lo} monocytes/macrophages (**Supplementary Fig. 5b**).

Question 10:

Supplementary Fig. 2, no data is presented for p<0.01.

Response:

Thanks for careful reading of the manuscript. We have deleted “p<0.01” in the legends

of the Figure.

Question 11:

Supplement, Fig 1 appears to show significant changes in immune cell populations in addition to macrophages, thus the basis for the author's statement that "other immune cell populations did not change significantly" (page 5) is not clear.

Response:

Thanks for your comments. Your concerns remind us that the description of the results is not accurate, so we changed the statement to "we found that monocytes/macrophages and neutrophils were the major cell types among the infiltrating cells".

Question 12:

Fig. 1, neutrophils may contribute to tissue repair, but there is no evidence that "neutrophil depletion delays...liver repair after acute injury" (legend to Fig. 1).

Response:

Thanks for your good suggestions. We changed the statement to "neutrophils contribute to tissue repair".

Question 13:

In chimera studies using Gcsf and Nox2 ko bone marrow, there was no consideration of the fact that there is partial depletion of macrophages and this can markedly alter the conclusions from these studies.

Response:

Thanks for your valuable comments. We acknowledge that there are pitfalls in chimera models due to the partial depletion of Nox2 in macrophages. Given the limited specificity of existing neutrophil-specific conditional knockout mice, we performed an adoptive transfer experiment to confirm the role of neutrophil-derived ROS, as mentioned in Response to **Question 1**.

Reference:

1. Passegue E, Wagner EF, Weissman IL. JunB deficiency leads to a myeloproliferative disorder arising from hematopoietic stem cells. *Cell* 2004, **119**(3): 431-443.
2. Lagasse E. bcl-2 inhibits apoptosis of neutrophils but not their engulfment by macrophages. *Journal of Experimental Medicine* 1994, **179**(3): 1047-1052.
3. Elliott ER, Van Ziffle JA, Scapini P, Sullivan BM, Locksley RM, Lowell CA. Deletion of Syk in neutrophils prevents immune complex arthritis. *Journal of immunology* 2011, **187**(8): 4319-4330.
4. Thomas RM, Schmedt C, Novelli M, Choi BK, Skok J, Tarakhovsky A, *et al.* C-terminal SRC kinase controls acute inflammation and granulocyte adhesion. *Immunity* 2004, **20**(2): 181-191.
5. Lim K, Hyun YM, Lambert-Emo K, Capece T, Bae S, Miller R, *et al.* Neutrophil trails guide influenza-specific CD8(+) T cells in the airways. *Science (New York, NY)* 2015, **349**(6252): aaa4352.
6. Van Ziffle JA, Lowell CA. Neutrophil-specific deletion of Syk kinase results in reduced host defense to bacterial infection. *Blood* 2009, **114**(23): 4871-4882.
7. Clausen BE, Burkhardt C, Reith W, Renkawitz R, Forster I. Conditional gene targeting in macrophages and granulocytes using LysMcre mice. *Transgenic research* 1999, **8**(4): 265-277.
8. Nemeth T, Mocsai A. The role of neutrophils in autoimmune diseases. *Immunology letters* 2012, **143**(1): 9-19.
9. Abram CL, Roberge GL, Hu Y, Lowell CA. Comparative analysis of the efficiency and specificity of myeloid-Cre deleting strains using ROSA-EYFP reporter mice. *Journal of immunological methods* 2014, **408**: 89-100.
10. Tkalcevic J, Novelli M, Phylactides M, Iredale JP, Segal AW, Roes J. Impaired immunity and enhanced resistance to endotoxin in the absence of neutrophil elastase and cathepsin G. *Immunity* 2000, **12**(2): 201-210.
11. Orthgiess J, Gericke M, Immig K, Schulz A, Hirrlinger J, Bechmann I, *et al.* Neurons exhibit Lyz2 promoter activity in vivo: Implications for using LysM-Cre mice in myeloid cell research. *European journal of immunology* 2016, **46**(6): 1529-1532.
12. Hagglof T, Sedimbi SK, Yates JL, Parsa R, Salas BH, Harris RA, *et al.* Neutrophils license iNKT cells to regulate self-reactive mouse B cell responses. *Nature immunology* 2016, **17**(12): 1407-1414.
13. Xiong H, Keith JW, Samilo DW, Carter RA, Leiner IM, Pamer EG. Innate Lymphocyte/Ly6C(hi) Monocyte Crosstalk Promotes Klebsiella Pneumoniae Clearance. *Cell* 2016, **165**(3): 679-689.
14. Halim TY, Hwang YY, Scanlon ST, Zaghoulani H, Garbi N, Fallon PG, *et al.* Group 2 innate lymphoid cells license dendritic cells to potentiate memory TH2 cell responses. *Nature immunology* 2016, **17**(1): 57-64.
15. Matsuda M, Tsurusaki S, Miyata N, Saijou E, Okochi H, Miyajima A, *et al.* Oncostatin M causes liver fibrosis by regulating cooperation between hepatic stellate cells and macrophages in mice. *Hepatology* 2018, **67**(1): 296-312.
16. Gabanyi I, Muller PA, Feighery L, Oliveira TY, Costa-Pinto FA, Mucida D. Neuro-immune Interactions Drive Tissue Programming in Intestinal Macrophages. *Cell* 2016, **164**(3): 378-391.

17. Xu X, Xu JF, Zheng G, Lu HW, Duan JL, Rui W, *et al.* CARD9(S12N) facilitates the production of IL-5 by alveolar macrophages for the induction of type 2 immune responses. *Nature immunology* 2018, **19**(6): 547-560.
18. Ramachandran P, Pellicoro A, Vernon MA, Boulter L, Aucott RL, Ali A, *et al.* Differential Ly-6C expression identifies the recruited macrophage phenotype, which orchestrates the regression of murine liver fibrosis. *Proceedings of the National Academy of Sciences of the United States of America* 2012, **109**(46): E3186-3195.

Reviewer #2 (ROS, neutrophil)(Remarks to the Author):

The authors addressed all of the comments of the reviewers and significantly improved the manuscript.

Response:

We recognize and deeply appreciate the effort and generosity that present in the constructive review. We wish to express our sincere thanks for your suggestions and comments, which are very valuable and helpful for revising and improving our manuscript.

Reviewer #3 (Neutrophil biology, liver inflammation)(Remarks to the Author):

This is an interesting manuscript with an interesting relationship between neutrophils and monocytes/macrophage in sterile injury and repair. I have a number of general comments that should be addressed.

Response:

First, we wish to express our sincere thanks for your suggestions and comments, which are very valuable and helpful for revising and improving our manuscript. Your questions and suggestions have been thoroughly discussed among all authors and we addressed your concerns in the following.

Question 1:

Not sure that a Ly6C high macrophage is an accepted term. Generally, these are referred to as monocytes. Throughout the manuscript the authors refer to Ly6C high and Ly6C low macrophage. However, immunologists would see this as markers of monocytes and macrophage respectively. While this could just be terminology it could also be a completely alternative interpretation ie., neutrophils help monocytes mature into macrophage. The authors must demonstrate at the beginning of their results that they truly are looking at macrophage and macrophage ie., is F4/80 and/or MerTK and/or other macrophage markers on the surface of both Ly6C high and Ly6C low cells.

Response:

Thanks for your valuable suggestions. We have assessed the expression of key macrophage markers F4/80, CD64, and the tyrosine kinase MerTK (**Supplementary Fig. 1b**). Both 24 hr Ly6C^{hi} cells and 72hr Ly6C^{lo} cells expressed macrophage markers F4/80, CD64 and MerTK, but the expression of these macrophage markers was lower in 24 hr Ly6C^{hi} cells than in 72 hr Ly6C^{lo} cells. These results suggest that Ly6C^{hi} cells may include infiltrating monocytes, monocytes in differentiation to macrophages and/or macrophages. Therefore, the identification of Ly6C^{hi} cells as macrophages in the previous manuscript is not accurate. We changed the description “Ly6C^{hi} macrophages” to “Ly6C^{hi} monocytes/macrophages” in the revision.

Question 2:

In the abstract and throughout the manuscript the authors make the following claim verbatim: “We mechanistically demonstrate that neutrophil-derived reactive oxygen species (ROS) trigger the phenotypic switch between macrophages via AMP-activated protein kinase (AMPK)” This is not shown in this manuscript. The authors show that AMPK is oxidized/activated but whether it is responsible in the switch is not examined. The authors only make one sentence statement saying that the knockout has fewer repair genes. This needs to be better fleshed out. Was there less repair in this knockouts?

Response:

Thanks for your good suggestions. We acknowledge that we did not provide sufficient evidence about the role of AMPK *in vivo*. Because other cell types in the liver^{1,2}, such as hepatocytes and hepatic stellate cells, are reported to express AMPK, AMPK systemic knockout mice may be not suitable for us to assess the role of AMPK in macrophages.

Since we could not obtain macrophage-specific conditional knockout mice in the short term, we examined the role of macrophagic AMPK by adoptive transfer of WT or AMPK $\alpha^{-/-}$ monocytes to *Ccr2^{-/-}* mice. Adoptive transfer of WT monocytes rescued the exacerbated damage and depressed hepatic regeneration in *Ccr2^{-/-}* mice during the resolution phase, whereas transfer of AMPK $\alpha^{-/-}$ monocytes failed to do so (**Fig. 6h-j**). These results may suggest that macrophagic AMPK is involved in liver repair. Given that our current results only suggest a possible mechanism by which ROS regulate macrophage transition and liver repair, we have tuned down our statements of relevant conclusions in the revised manuscript.

Question 3:

I am not sure the results of depletion of neutrophils in CCR2 $^{-/-}$ mice provide a clear conclusion that neutrophil repair is related to monocytes.

Response:

Thanks for your valuable comments. In the previous manuscript, we showed that mice

lacking neutrophils displayed exacerbated damage and depressed hepatic regeneration during the resolution phase, but this effect was lost upon monocyte/macrophage ablation in *Ccr2*^{-/-} mice. Thus, we speculated that neutrophils mediate liver repair may be related to monocytes/macrophages. However, given that *Ccr2*^{-/-} mice already resulted in increased liver damage, it may not be appropriate to see if neutrophil depletion would cause a worsen effect in these mice. Based on this, we performed an adoptive transfer experiment to investigate whether neutrophils could alleviate liver damage in WT or *Ccr2*^{-/-} mice. We found that adoptive transfer of neutrophils alleviated liver damage in WT mice during the resolution phase (**below Figures a-c**), whereas transfer of neutrophils had no effect in *Ccr2*^{-/-} mice (**below Figures d-f**). These results may indicate that the neutrophil function in liver repair requires the presence of monocytes/ macrophages.

Reference:

1. Caligiuri A, Bertolani C, Guerra CT, Aleffi S, Galastri S, Trappoliere M, *et al.* Adenosine monophosphate-activated protein kinase modulates the activated phenotype of hepatic stellate cells. *Hepatology* 2008, **47**(2): 668-676.

2. Vazquez-Chantada M, Ariz U, Varela-Rey M, Embade N, Martinez-Lopez N, Fernandez-Ramos D, *et al.* Evidence for LKB1/AMP-activated protein kinase/ endothelial nitric oxide synthase cascade regulated by hepatocyte growth factor, S-adenosylmethionine, and nitric oxide in hepatocyte proliferation. *Hepatology* 2009, **49**(2): 608-617.

Reviewers' comments:

Reviewer #1 (Remarks to the Author):

1. Data does not strongly support the idea that neutrophil-derived ROS mediate effects on macrophages. The title should be changed to more accurately represent what the data actually shows...perhaps simply "Evidence that neutrophils regulate macrophage activity..." without reference to 'licensing?' or that the process is mediated by "ROS"; also eliminate references to ROS in the Introduction. There could be a separate section in the 'Results' asking the question of whether ROS mediate the effects of neutrophils, data on ROS could then be presented with a detailed discussion of the hypothesis and limitations of the data as well as the fact that neutrophils release many other mediators in addition to ROS. The Discussion should include a detailed review of the literature comparing our current understanding of the roles of neutrophils and macrophages in toxicity.
2. There is a disconnect between the timing of when neutrophils accumulate in the liver (and produce ROS) following exposure to a toxicant (first 24 hours) and when there are alterations in macrophage functioning (at much later time points). This should be addressed in the Discussion. This reviewer notes that neutrophil-derived ROS could trigger a response after many hours and this should be addressed in the Discussion.

Reviewer #3 (Remarks to the Author):

The manuscript is greatly improved and the authors have made a genuine effort to revise appropriately. I have two very small changes the authors may wish to consider.

- 1) As I mentioned previously, and the authors have taken under advisement they are watching monocytes convert to repair macrophage. The authors changed the word macrophage to monocyte/macrophage everywhere except the title. I am not going to force them to change their title but wonder whether the title reflects the manuscript.
- 2) I completely forgot that NAC was used to treat patients with APAP poisoning. As such calling it an anti-oxidant as the authors do in the results, undermines their very nice paper. They may wish to add a sentence saying that NAC may have other effects in the results section.

Point-by-point Response to Reviewers

Reviewer comments:

Reviewer #1 (Remarks to the Author):

Question1:

Data does not strongly support the idea that neutrophil-derived ROS mediate effects on macrophages. The title should be changed to more accurately represent what the data actually shows...perhaps simply "Evidence that neutrophils regulate macrophage activity...." without reference to 'licensing?' or that the process is mediated by "ROS"; also eliminate references to ROS in the Introduction. There could be a separate section in the 'Results' asking the question of whether ROS mediate the effects of neutrophils, data on ROS could then be presented with a detailed discussion of the hypothesis and limitations of the data as well as the fact that neutrophils release many other mediators in addition to ROS. The Discussion should include a detailed review of the literature comparing our current understanding of the roles of neutrophils and macrophages in toxicity.

Response:

First, we wish to express our sincere thanks for your suggestions and comments, which are very valuable and helpful for revising and improving our manuscript. We recognize and deeply appreciate the effort and generosity that present in your constructive review.

Your comments and suggestions have been thoroughly discussed among all authors. We realized that our previous statements about “neutrophil-derived ROS” may be misleading because we could not rule out that additional factors other than ROS might contribute to macrophage skewing and liver repair. To be more precise, we concluded that “our results suggested that neutrophils trigger macrophage skewing toward a reparative phenotype for optimal liver repair and the process is mediated by ROS”.

In addition, we summarized several evidences on the role of ROS in macrophage conversion and liver repair, as shown below. We believe these findings could support the above conclusions and may be worth reconsidering.

(1) We found that ROS deficient *Nox2*^{-/-} mice displayed delayed resolution of liver injury and impaired macrophage conversion.

(2) Fate mapping of *Cx3cr1*⁺Ly6C^{hi} monocytes/macrophages in *Cx3cr1*^{CreERT2-EYFP/+}*Rosa26*^{tdTomato/+} mice, we found that significant lower proportion of the tdTomato⁺ macrophages converted to the Ly6C^{lo} phenotype in the ROS-blocked mice.

(3) Using mixed bone marrow chimeras, we found that *Nox2*^{-/-}/*Gcsf*^{-/-} mixed bone marrow chimeric mice profoundly impaired liver regeneration and macrophage conversion during the resolution phase, as compared to WT/*Gcsf*^{-/-} chimeras.

(4) Using neutrophil adoptive transfer systems, we found that adoptive transfer of WT neutrophils rescued the exacerbated damage, depressed hepatic regeneration and impaired macrophage skewing in neutrophil-depleted mice during the resolution phase, whereas transfer of *Nox2*^{-/-} neutrophils failed to do so.

(5) Using co-culture systems, we found that conditioned medium from WT neutrophils, but not *Nox2*^{-/-} neutrophils, promoted pro-inflammatory monocytes/macrophages skewing toward a reparative phenotype.

Although our results suggest that ROS play important roles in regulating macrophage skewing and liver repair, as you mentioned, we cannot exclude that other mediators released by neutrophils may have similar effects as ROS. Therefore, in order to present our results more objectively and accurately, we made the following changes in the writing of the manuscript according to your suggestions.

(1) Regarding the title and introduction, as you suggested, changing the title to “Neutrophils promote the development of reparative macrophages to orchestrate liver repair and the process is mediated by ROS” will more accurately represent our results. Given the limitation of the title length in the manuscript preparation guidelines, we

finally changed the title to “Neutrophils promote the development of reparative macrophages mediated by ROS to orchestrate liver repair”. Additionally, for a more accurate summary in Introduction, we changed the related statements to “We show that neutrophils instruct inflammatory monocytes/macrophages to adopt a pro-regenerative phenotype for optimal liver repair and the process is mediated by ROS”.

(2) Regarding the Results section about ROS, we changed the statement of our hypothesis to “Because activated neutrophils can potently produce ROS and phagocyte NADPH oxidase 2 (Nox2) complex-derived ROS are known to be crucial regulators of immune response, we asked whether ROS might mediate the regulation of the macrophage skewing”. Importantly, we added a detailed discussion of the hypothesis and limitations of the data: “However, we cannot exclude that additional factors other than ROS might contribute to macrophage skewing and liver repair. Since neutrophils are reported to generate many anti-inflammatory and pro-resolving mediators, whether these products have similar effects remains to be investigated. Nevertheless, we identified ROS expressed predominantly by neutrophils are important regulators of macrophage skewing for liver repair.”

(3) Regarding the Discussion section, we acknowledged that our previous manuscript did lack a detailed review about the roles of neutrophils and macrophages in APAP-induced liver injury. According to your suggestions, we have added the related discussion as below: “Previous studies have shown that neutrophils and distinct monocyte/macrophage subsets accumulated in the liver after APAP-induced injury^{1,2}. The diverse roles of two different infiltrated monocyte/macrophage populations have been extensively studied³⁻⁷. Ly6C^{hi}CX₃CR1^{lo} monocytes/macrophages exhibited a pro-inflammatory phenotype and mediated the development of liver injury, while Ly6C^{lo}CX₃CR1^{hi} macrophages dampened inflammation and promoted tissue repair. However, the in vivo function of neutrophils has been less well characterized, compared to that of macrophages. Increasing amounts of data indicate that neutrophils do not contribute to the liver injury during the early phase after APAP overdose⁸. Nevertheless, the role of

neutrophils during the resolution phase remains largely unexplained. Here, we provide evidence that neutrophils are required for optimal liver regeneration and repair through promoting macrophage conversion.”

The following related literatures were cited:

- [1] Liu ZX, Govindarajan S, Kaplowitz N. Innate immune system plays a critical role in determining the progression and severity of acetaminophen hepatotoxicity. *Gastroenterology* 2004, 127(6): 1760-1774.
- [2] Dambach DM, Watson LM, Gray KR, Durham SK, Laskin DL. Role of CCR2 in macrophage migration into the liver during acetaminophen-induced hepatotoxicity in the mouse. *Hepatology* 2002, 35(5): 1093-1103.
- [3] Laskin DL, Sunil VR, Gardner CR, Laskin JD. Macrophages and tissue injury: agents of defense or destruction? *Annual review of pharmacology and toxicology* 2011, 51: 267-288.
- [4] Dragomir AC, Sun R, Choi H, Laskin JD, Laskin DL. Role of galectin-3 in classical and alternative macrophage activation in the liver following acetaminophen intoxication. *Journal of immunology* 2012, 189(12): 5934-5941.
- [5] Mossanen JC, Krenkel O, Ergen C, Govaere O, Liepelt A, Puengel T, et al. Chemokine (C-C motif) receptor 2-positive monocytes aggravate the early phase of acetaminophen-induced acute liver injury. *Hepatology* 2016, 64(5): 1667-1682.
- [6] Laskin DL, Laskin JD. Role of macrophages and inflammatory mediators in chemically induced toxicity. *Toxicology* 2001, 160(1-3): 111-118.
- [7] Gardner CR, Hankey P, Mishin V, Francis M, Yu S, Laskin JD, et al. Regulation of alternative macrophage activation in the liver following acetaminophen intoxication by stem cell-derived tyrosine kinase. *Toxicology and applied pharmacology* 2012, 262(2): 139-148.
- [8] Jaeschke H, Williams CD, Ramachandran A, Bajt ML. Acetaminophen hepatotoxicity and repair: the role of sterile inflammation and innate immunity. *Liver international : official journal of the International Association for the Study of the Liver* 2012, 32(1): 8-20.

Question 2:

There is a disconnect between the timing of when neutrophils accumulate in the liver (and produce ROS) following exposure to a toxicant (first 24 hours) and when there are alterations in macrophage functioning (at much later time points). This should be addressed in the Discussion. This reviewer notes that neutrophil-derived ROS could trigger a response after many hours and this should be addressed in the Discussion.

Response:

This is an interesting question. This may be due to the fact that it takes time for

macrophages to alter their phenotypes in response to local tissue signals. Several studies have shown that macrophages require time for their phenotypic conversion. For example, in a mouse model of peritoneal inflammation, it took 8 weeks for recruited CX₃CR1⁺ monocyte-derived cells to adopt a tissue-resident peritoneal macrophage phenotype¹. In a thermal-induced liver injury, CCR2^{hi}CX₃CR1^{lo} monocytes were recruited to the injured area at 24 h and transitioned into CCR2^{lo}CX₃CR1^{hi} monocytes at 48 h². Thus, in APAP-induced liver injury, monocytes/ macrophages may also need some time to switch their phenotypes. In addition, consistent with previous reports³, we found that neutrophils released ROS continuously throughout the time course after APAP overdose (Fig.3a and Supplementary Fig.4b), which could maintain the local microenvironment that may regulate macrophage conversion.

Reference:

1. Gundra UM, Girgis NM, Gonzalez MA, San Tang M, Van Der Zande HJP, Lin JD, *et al.* Vitamin A mediates conversion of monocyte-derived macrophages into tissue-resident macrophages during alternative activation. *Nature immunology* 2017, **18**(6): 642-653.
2. Dal-Secco D, Wang J, Zeng Z, Kolaczowska E, Wong CH, Petri B, *et al.* A dynamic spectrum of monocytes arising from the in situ reprogramming of CCR2+ monocytes at a site of sterile injury. *The Journal of experimental medicine* 2015, **212**(4): 447-456.
3. Williams CD, Bajt ML, Sharpe MR, McGill MR, Farhood A, Jaeschke H. Neutrophil activation during acetaminophen hepatotoxicity and repair in mice and humans. *Toxicology and applied pharmacology* 2014, **275**(2): 122-133.

Reviewer #3 (Remarks to the Author):

The manuscript is greatly improved and the authors have made a genuine effort to revise appropriately. I have two very small changes the authors may wish to consider.

Response:

First, we wish to express our sincere thanks for your suggestions and comments, which are very valuable and helpful for revising and improving our manuscript. We recognize and deeply appreciate the effort and generosity that present in your constructive review.

Question 1:

As I mentioned previously, and the authors have taken under advisement they are watching monocytes convert to repair macrophage. The authors changed the word macrophage to monocyte/macrophage everywhere except the title. I am not going to force them to change their title but wonder whether the title reflects the manuscript.

Response:

Thanks for your good suggestions. To make the title more accurately represent the manuscript, we have changed the title to “Neutrophils promote the development of reparative macrophages mediated by ROS to orchestrate liver repair”.

Question 2:

I completely forgot that NAC was used to treat patients with APAP poisoning. As such calling it an anti-oxidant as the authors do in the results, undermines their very nice paper. They may wish to add a sentence saying that NAC may have other effects in the results section.

Response:

Thanks for your good suggestions. We’re sorry that we did not explicitly explain the issue of NAC in the previous Point-by-point response, which make you feel confused. NAC is not only an inhibitor of ROS, but also an antidote in acetaminophen

intoxication.

However, the effect of NAC we observed during the resolution phase may be independent of its detoxification mechanism. There are two main reasons:

(1) The detoxification mechanism of NAC is only effective during the early phase following APAP overdose. In the clinical setting, NAC is effective only for patients who present within eight hours of an acute overdose, and is less effective for late-presenting patients^{1,2,3}. In mice, NAC only prevented toxicity when administered prior to or early following APAP⁴.

(2) Late or prolonged treatment with NAC is indeed toxic (detrimental) rather than detoxification (protective) in mice. Several studies in mice have reported that late or prolonged NAC treatment clearly decreased survival and impaired liver regeneration^{5,6}. In our previous data, two doses of NAC were injected at 0 and 36 h to induce sustained ROS scavenging in APAP-challenged mice. We found that although NAC resulted in alleviated liver damage during the initial phase of inflammation, it exacerbated injury and impaired regeneration during the resolution phase of inflammation, which was consistent with previous results⁶. The increased severity of APAP-induced liver injury during the resolution phase could not be explained by APAP detoxification, but may be due to ROS scavenging.

We also have added a brief explanation about the effects of NAC in the Results section as below: “It is worth noting that although NAC also acts as an antidote in APAP intoxication, the effect of NAC we observed during the resolution phase may be independent of its detoxification mechanism. There are two main reasons: Firstly, the detoxification mechanism of NAC is only effective during the early phase following APAP overdose in humans and mice. Secondly, Late or prolonged treatment with NAC is indeed detrimental rather than protective in mice, which could not be explained by APAP detoxification.”

Reference:

1. Smilkstein MJ, Knapp GL, Kulig KW, Rumack BH. Efficacy of oral N-acetylcysteine in the

treatment of acetaminophen overdose. Analysis of the national multicenter study (1976 to 1985). *The New England journal of medicine* 1988, **319**(24): 1557-1562.

2. Kerr F, Dawson A, Whyte IM, Buckley N, Murray L, Graudins A, *et al.* The Australasian Clinical Toxicology Investigators Collaboration randomized trial of different loading infusion rates of N-acetylcysteine. *Annals of emergency medicine* 2005, **45**(4): 402-408.
3. Whyte IM, Francis B, Dawson AH. Safety and efficacy of intravenous N-acetylcysteine for acetaminophen overdose: analysis of the Hunter Area Toxicology Service (HATS) database. *Current medical research and opinion* 2007, **23**(10): 2359-2368.
4. James LP, McCullough SS, Lamps LW, Hinson JA. Effect of N-acetylcysteine on acetaminophen toxicity in mice: relationship to reactive nitrogen and cytokine formation. *Toxicological sciences : an official journal of the Society of Toxicology* 2003, **75**(2): 458-467.
5. Yang R, Miki K, He X, Killeen ME, Fink MP. Prolonged treatment with N-acetylcysteine delays liver recovery from acetaminophen hepatotoxicity. *Critical care* 2009, **13**(2): R55.
6. Banda PW, Quart BD. The use of N-acetylcysteine long after an acetaminophen overdose in mice. *Toxicology letters* 1987, **36**(1): 89-94.

REVIEWERS' COMMENTS:

Reviewer #1 (Remarks to the Author):

The authors addressed all comments. A very good case was made to support the idea that neutrophils have the capacity to alter macrophage phenotype.